



# A Novel Model Hierarchy Isolates the Effect of Temperature-dependent Cloud Optics on Infrared Radiation

Ash Gilbert[1,2], Jennifer E. Kay[1,2], and Penny Rowe[3]

[1]Department of Atmospheric and Oceanic Sciences, University of Colorado Boulder, Boulder, CO, USA
[2]Cooperative Institute for Research in Environmental Sciences, University of Colorado Boulder, Boulder, CO, USA
[3]NorthWest Research Associates, Seattle, WA, USA

**Correspondence:** Ash Gilbert (ash.gilbert@colorado.edu)

**Abstract.** Clouds exert strong influences on surface energy budgets and climate projections. Yet, cloud physics is complex and often incompletely represented in models. For example, temperature-dependent cloud optics parameterizations are rarely incorporated into the radiative transfer models used for future climate projections. Prior work has shown that incorporating these optics in downwelling longwave radiation calculations results in increases of as much as 1.7 W m$^{-2}$ for Arctic atmospheres.
Here we examine whether implementing these optics in climate models leads to significant climate impacts. We use a novel methodology based on a hierarchy of models. In two-stream radiation and single-column models, incorporating temperature-dependent optical properties had a small impact (< 1 W m$^{-2}$). Similarly, impacts were statistically insignificant on infrared radiation within freely evolving atmospheric model simulations. In contrast, there was a much larger effect (1–3 W m$^{-2}$) from optics changes when the winds within our atmospheric model experiments were nudged towards reanalysis winds. This new application of wind-nudging experiments helped to isolate the effect from temperature-dependent cloud optics changes by reducing the internally generated atmospheric variability. In summary, we found a signal from temperature-dependent optics, but this effect is small compared to climate variability and didn't impact long term Arctic temperature trends. More broadly, this work demonstrates a new framework for assessing the climate importance of a physics change.

## 1 Introduction

Due to complex processes that couple cloud processes with the climate system, cloud radiative effects remain one of the largest sources of climate projection uncertainty (Webb et al., 2017; Sherwood et al., 2020). Clouds affect climate by absorbing and emitting longwave radiation and scattering shortwave radiation. The strength of these cloud radiative effects depends on the cloud properties, including the phase, particle size and number, and geometric thickness. For example, optically thick liquid clouds scatter more shortwave radiation and emit more longwave radiation than optically thin ice clouds. All else being equal, clouds with small particle sizes also scatter more shortwave and emit more downwelling longwave than clouds with large particle sizes.

In most climate models, translating cloud properties into cloud radiative impacts is accomplished through cloud optics parameterizations. Using an appropriate level of complexity in cloud optics parameterizations is therefore critical to accurately modeling cloud radiative impacts. Due to the computational expense of radiative transfer calculations, choices must be made



about what aspects of cloud optics are incorporated. These choices should be re-assessed to include new physics when the impacts on radiation are substantial. Developing tools to assess whether a cloud optics change substantially affects model radiative fluxes is therefore of practical importance to the model development community.

A useful yet underutilized technique for isolating the importance of a cloud optics change for climate is wind nudging. In this technique, model winds are nudged towards prescribed wind values, often observed or reanalysis data, over a set horizontal
and vertical domain. The value of nudging the winds to the prescribed values is that the time evolution of the prescribed and modeled large-scale circulation is synchronized. Typically, winds are nudged above the boundary layer, leaving the boundary layer physics, including the surface fluxes and low clouds, to evolve interdependently. Recent applications show the power of wind nudging for scientific and direct model comparisons with observations. For example, Pithan et al. (2023) compared nudged model runs to observations and made specific suggestions for model microphysical parameterization improvements
in the Arctic. Likewise, Kooperman et al. (2012) leveraged the synchronizing of large-scale wind evolution enabled by wind nudging to increase the detectability of an aerosol radiative signal. These studies show that wind nudging is a powerful tool for highlighting non-dynamical signals by constraining the atmospheric circulation in a climate model.

Based on these previous studies, what knowledge gaps does this study want to address? We identify a cloud optics physics that has not been incorporated into the radiation scheme used by many climate models, RRTMG (Iacono et al., 2008). Specif-
ically, temperature-dependent liquid water optics are not used in RRTMG. Yet, using high-spectral resolution models applied to case studies in Antarctica, Rowe et al. (2013) found that these optics can change longwave fluxes emitted by supercooled liquid-containing clouds by up to $1.7 \ \mathrm{W \ m^{-2}}$. Here, we assess if using temperature-dependent liquid water optics for downwelling longwave radiation in the Arctic has substantial impacts on radiation in a global climate context. Thus, a primary goal of this study is to assess if this cloud optics change should be considered as a candidate for addition to the RRTMG radia-
tive transfer model used by most climate models. Our assessment will focus on the Arctic, where supercooled liquid clouds frequently occur in both observations (Cesana et al., 2012) and the climate model we use (e.g. Community Earth System Model Version 2) (McIlhattan et al., 2020) and where the atmosphere is typically cold and dry, allowing significant infrared downwelling radiance from clouds to reach the surface (Rowe et al., 2013).

A novel aspect of this study is using a hierarchy of models to assess the relevance of a cloud optics change. From simplest to
most complex, these models are a two-stream radiative transfer model, a single-column atmospheric model, a freely evolving global climate model, and a wind-nudged global climate model. For each model, we then assess changes in longwave radiation produced by the temperature-dependent optics change that we make. While this study focuses on one specific optics change, the novel hierarchy and methods used here are applicable to any cloud optics change and therefore should be of broad interest to the model development community. Thus, a secondary goal is to establish the utility of this novel model hierarchy for assessing
the importance of a physics parameterization change for the climate.





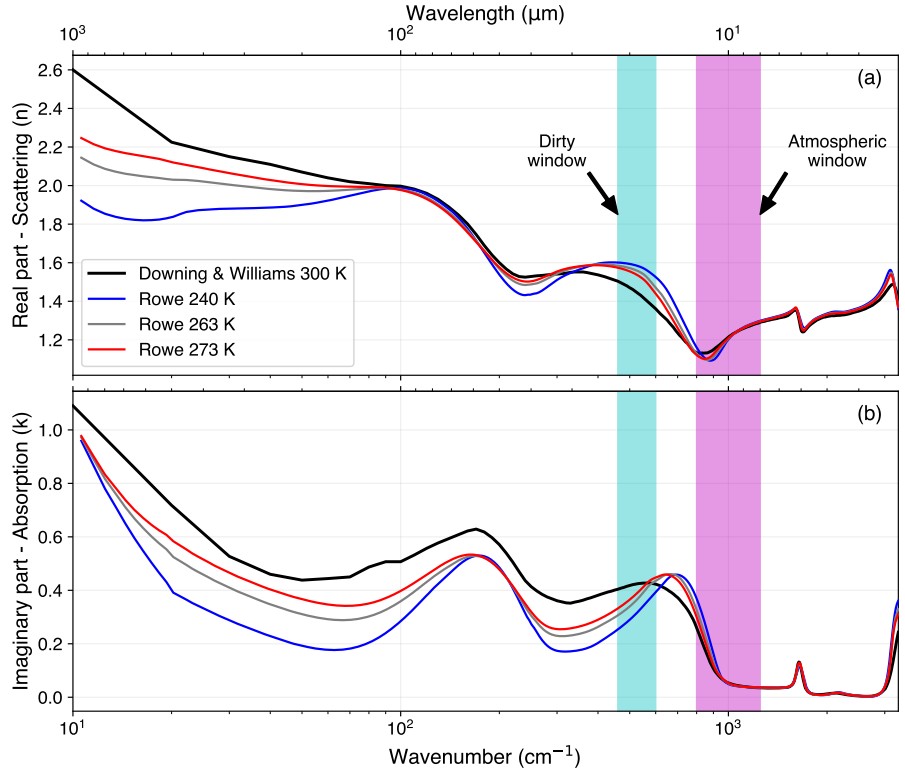

**Figure 1.** Complex refractive index of water with the (a) real part and (b) imaginary part between 10 and 3000 cm$^{-1}$ for four complex refractive indices at different temperatures: 240 K, 263 K, and 273 K from Rowe et al. (2020) and 300 K from Downing and Williams (1975). The highlighted regions are the dirty (blue) and atmospheric (pink) windows.

## 2 Methods

### 2.1 Temperature-dependent liquid water optics

Our cloud optics change is the implementation of temperature-dependent liquid water optics relevant for supercooled liquid water. Supercooled liquid water (240–273 K) scatters and absorbs radiation differently than room temperature water ($\sim 298$ K) (Rowe et al., 2020). Rowe et al. (2013) focused on the consequences of including the temperature dependence of liquid cloud optics in two spectral regions where laboratory measurements are sufficiently accurate to determine the temperature dependence of refractive indices of supercooled liquid and where clouds have a strong impact on downwelling longwave radiation, in the dirty window (460–640 cm$^{-1}$) and in part of the atmospheric window (760–990 cm$^{-1}$). We define optics that account for this temperature dependence as "temperature-dependent" and optics that assume all water behaves the same as room temperature water as "temperature-independent". Figure 1 illustrates the difference in optics by plotting temperature-dependent and temperature-independent complex refractive indices, which are defined as how a given material scatters and





absorbs radiation as a function of wavelength. Optical properties used in our study were derived from these complex refractive indices. For the temperature-dependent optics, we used complex refractive indices from Rowe et al. (2020) at the temperatures of 240 K, 253 K, 263 K, and 273 K.

## 2.2 Model hierarchy

In this work, we evaluate the effect of changing the liquid water optics from temperature-independent to temperature-dependent on longwave radiation at different scales. Therefore, we developed a model hierarchy with increasing complexity that includes four models:

1. **Two-stream radiative transfer model:** a simplistic radiative transfer model that simulates the downwelling longwave spectra from a single supercooled liquid cloud. *Do we see an effect with a simple mathematical model on a spectral scale?*

2. **Single-column atmospheric model:** a single grid point version of a global atmospheric model constrained by observations for one month. *Do we see an effect with an atmospheric model at a single location on a daily time scale?*

3. **Freely evolving global climate model:** a climate model run over the entire globe for several decades. *Do we see an effect with a global climate model over the entire Arctic on a decadal time scale?*

4. **Wind-nudged global climate model:** a climate model run over the entire globe for a single year and for several decades with the winds constrained to enhance the signal in the radiation. *Do we see an effect with a dynamically constrained global climate model over the entire Arctic on annual and decadal time scales?*

For each model, we compared the longwave radiation produced using temperature-independent optics against longwave radiation produced using temperature-dependent optics. Then, we evaluated whether the difference in radiation was detectable and statistically significant. Finally, we assessed at what scales and for which models the temperature dependence of liquid water optics mattered.

## 2.3 Two-stream radiative transfer model

The first model in our hierarchy is a two-stream radiative transfer model following the work of Petty (2006b) and Rowe et al. (2013) (Fig. 2). We modeled a single liquid cloud with a droplet number concentration ($N = 40 \ \mathrm{cm}^{-1}$) and an effective radius ($r_{\mathrm{eff}} = 10 \ \mu\mathrm{m}$) based on observations of mixed-phase clouds from Klein et al. (2009). We used a non-black surface with an albedo ($r_{\mathrm{sfc}} = 0.2$) and temperature ($T_{\mathrm{sfc}} = 250 \ \mathrm{K}$). For the temperature-dependent optics, we used the 240 K, 253 K, 263 K, and 273 K optics from Rowe et al. (2020). For the temperature-independent optics, we used the 300 K optics from Downing and Williams (1975). We calculated the Mie scattering properties (i.e. the asymmetry parameter, $g$; the extinction efficiency $Q_{\mathrm{ext}}$; and the scattering efficiency, $Q_{\mathrm{sca}}$) using the method outlined in Wiscombe (1979). The rest of the optical properties for the two-stream model we calculated using equations described in Petty (2006a). The full list of equations we used in the two-stream model is included in Appendix A1. We modeled the downwelling longwave spectra from a single supercooled



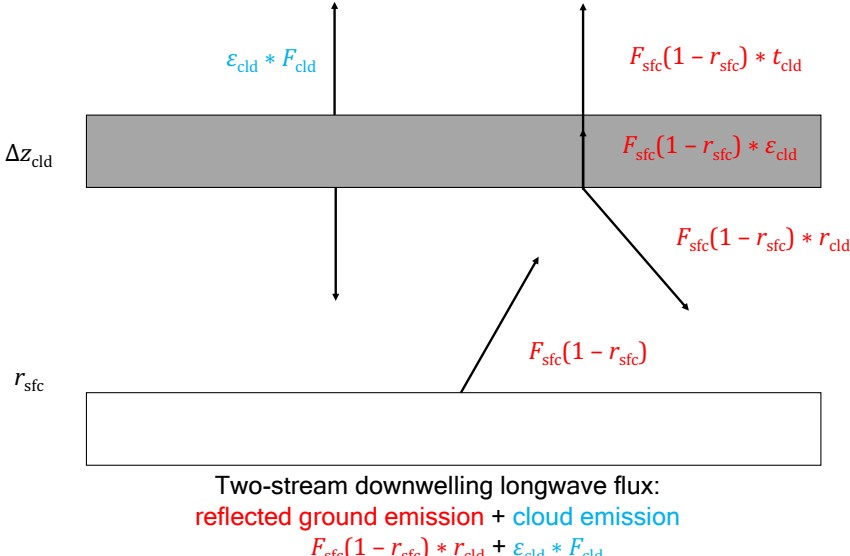

**Figure 2.** Diagram of the two-stream radiative transfer model as described in Petty (2006b). The gray layer represents the liquid cloud and the white layer the snow-covered ground. The cloud has a thickness $\Delta z_{cld}$. All quantities are functions of wavenumber. Quantities are color-coded by their emission source: red indicates original emission form the surface and blue from the cloud. $F_{sfc}$ and $F_{cld}$ represent the Planck black-body emissions of the snow surface and liquid cloud, respectively. $\epsilon_{cld}$ and $t_{cld}$ represent the emissivity and transmissivity of the liquid cloud, respectively. The reflectivity or albedo of the ground and cloud are represented by $r_{sfc}$ and $r_{cld}$, respectively.

liquid cloud over the wavenumbers 770-1000 $cm^{-1}$. We also modeled the cloud with three different thicknesses ($\Delta z_{cld}$: 100 m, 500 m, and 1000 m) and four different temperatures ($T_{cld}$: 240 K, 253 K, 263 K, and 273 K). For each temperature, we calculated the spectra once using the temperature-dependent optics that matched the cloud temperature and once using the temperature-independent optics.

## 2.4 Single-column atmospheric model

Next, we evaluated the impact of temperature-dependent optics within a single column model. Specifically, we used the Single-Column Atmospheric Model Version 6 (SCAM, Gettelman et al. (2019)), available as part of the Community Earth System Model Version 2 (CESM2) (Danabasoglu et al., 2020). SCAM has all of the physics of the atmospheric component of CESM2, the Community Atmosphere Model Version 6 (CAM), but only runs at a single location. We forced SCAM with 17 days of observations from the Mixed-Phase Arctic Cloud Experiment (MPACE) to simulate an Arctic atmosphere with mixed-phase and supercooled liquid-containing clouds (Harrington and Verlinde, 2005). In order to use our temperature-dependent optics in SCAM, we had to reproduce optics files in the same format as in SCAM/CESM2. The full description of how we created the optics files can be found in Appendix A2. We produced optics files for 240 K, 253 K, 263 K, and 273 K. The default liquid water optics file in CESM2 was based on 298 K water and we used these optics in all of our control experiments. SCAM and





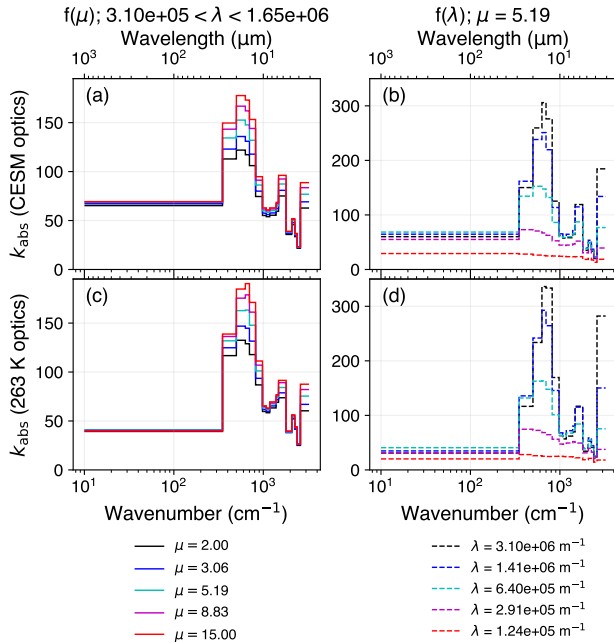

**Figure 3.** The longwave mass absorption coefficient ($k_{abs}$ ($m^2$ $kg^{-1}$)) graphed for the current CESM2 liquid optical properties (a) & (b) and for new liquid optical properties calculated from the 263 K complex refractive index (Rowe et al., 2020) (c) & (d) as a function of wavenumber and wavelength. In CESM2, $k_{abs}$ is a lookup table in terms of the parameters $\mu$ and $1/\lambda$ that describe the droplet size distribution where $\lambda$ is a function of $\mu$. (b) and (d) are the $k_{abs}$ spectra at a fixed $\mu$ and five $\lambda$. (a) and (c) are the $k_{abs}$ spectra at five $\mu$ and their corresponding $\lambda$.

CESM2 only use the mass absorption coefficient ($k_{abs}$) in the longwave, which is plotted for the default optics and one of the temperature-dependent optics sets, 263 K, in Fig. 3. For our temperature-dependent optics experiments, we swapped the default file for one of our temperature-dependent optics files such that any liquid water in the atmosphere has those properties.

To swap optics files, we gave SCAM the CAM namelist argument 'liqopticsfile' the file path to a temperature-dependent optics file. We did this namelist change for all SCAM and CESM2 simulations where we used temperature-dependent optics. We ran SCAM forced by MPACE with four sets of optics: the control optics and the temperature-dependent 240 K, 263 K, and 273 K optics. We chose these optics sets to mirror the sets we used in the freely evolving and wind-nudged global climate model experiments. We used the downwelling longwave flux at the surface (W $m^{-2}$) to evaluate if the optics changed how much

radiation the clouds emitted and absorbed. We used this variable for the rest of the model runs in CESM2.

## 2.5 Freely evolving global climate model

Next, we wanted to see the effect of temperature-dependent optics in a global climate on a large spatial scale and decadal temporal scale. We used the Community Earth System Model Version 2.2 (CESM2) for all our global climate model runs (Danabasoglu et al., 2020). We selected this climate model because it is a widely used, publicly available, and observationally



**Table 1.** CESM2 experiments list

| Experiment name | Compset | Duration | Ensemble members | Optics sets | Wind nudging |
|---|---|---|---|---|---|
| F1850 | F1850[a] | 40 years | 1 member | Control, 240 K, 273 K | – |
| F1850_UVnudge1980 | F1850 | 1 year | 10 members | Control, 240 K, 263 K, 273K | 67.5–82.5° N[c], above 820 hPa; U & V from ERA-I 1980 |
| B1850_UVnudge1980 | B1850[b] | 1 year | 10 members | Control, 263 K | 67.5–82.5° N, above 820 hPa; U & V from ERA-I 1980 |
| F1850_UVnudge1980–2018 | F1850 | 39 years | 3 members | Control, 263 K | 67.5–82.5° N, above 820 hPa; U & V from ERA-I 1980–2018 |

[a]Freely evolving atmosphere and land components with pre-industrial prescribed sea ice extent and ocean surface temperatures, greenhouse gas forcing, and initial conditions.
[b]Fully coupled model with pre-industrial greenhouse gas forcing and initial conditions.
[c]The nudging window doesn't cover the entire Arctic (60–90° N), but we conducted nudging window testing that shows little difference in the modeled radiation between the 67.5–82.5° N and the 60–90° N windows.

vetted climate models with wind nudging capabilities (Kooperman et al., 2012). Previous work has analyzed and exposed important CESM2 Arctic biases, including an overestimation of cloud liquid (McIlhattan et al., 2020) and insufficient late summer Arctic sea ice cover (DuVivier et al., 2020). Understanding these known biases is valuable for the work here. Notably, the overestimation of cloud liquid may amplify any effect of the temperature-dependent optics. In our model runs, we used a pre-industrial climate to examine the effect of temperature-dependent optics on the mean state of the Arctic climate. All

simulations had a spatial resolution of 1°x1°. For the first set of experiments, we ran three 20 year simulations of CESM2 with prescribed sea ice and ocean surface where each run had a different set of optics: control, 240 K, and 273 K. We chose the 240 K and 273 K optics because these temperatures are the outer limits for supercooled liquid water. This group of experiments was called F1850 and is described in Table 1.

## 2.6 Wind-nudged global climate model

For our next set of experiments, we use wind nudging, where the model uses a relaxation tendency term to nudge model values toward target values (Kooperman et al., 2012; Pithan et al., 2023). Nudging is implemented following:

$$\frac{\mathrm{d}x}{\mathrm{d}t} = F(x) + F_{\mathrm{nudge}}, \tag{1}$$

$$F_{\mathrm{nudge}} = \alpha \left[ O(t'_{\mathrm{next}}) - x(t) \right] / \tau, \tag{2}$$

where $F(x)$ the internal tendency without nudging, $F_{\mathrm{nudge}}$ is the nudging term, $\alpha$ is the strength coefficient that is 0 where

nudging is not enabled and 1 where nudging is enabled, $O(t'_{\mathrm{next}})$ is the model state at a future time step, $x(t)$ is the model state at the current time step, and $\tau$ is the time between the next time step and the current model step (Blanchard-Wrigglesworth et al., 2021; Roach and Blanchard-Wrigglesworth, 2022). In our experiments, we nudged the horizontal wind components



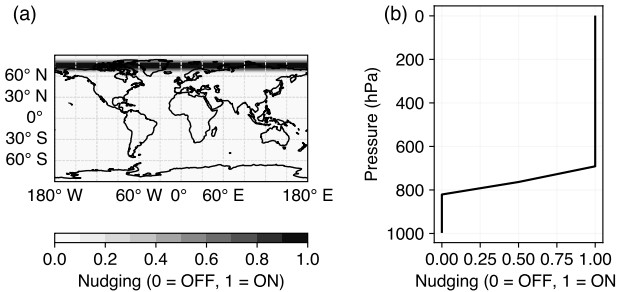

**Figure 4.** (a) Horizontal wind nudging window with nudging is on between 67.5-82.5° N. (b) Vertical wind nudging window with nudging is on above 820 hPa. Where wind nudging is enabled, the model nudges the horizontal wind components toward ERA-Interim reanalysis values.

of CESM2 between 67.5–82.5° N and above 820 hPa (Fig. 4). We nudged the model with 6-hourly ERA-Interim reanalysis (ERA-I) data (European Centre for Medium-Range Weather Forecasts, 2009).

All of our wind nudging experiments are detailed in Table 1. The first nudging experiment was a 1 year 10 member ensemble with prescribed sea ice and ocean and the winds nudged to 1980 values from ERA-I, called F1850. Nudging the winds constrains, but doesn't eliminate, the internal variability of the modeled climate system. Therefore, we ran this ensemble to quantify the internal variability of the wind-nudged climate system. We also ran the same 1 year 10 member wind-nudged ensemble with a fully coupled model in a set of simulations called B1850_UVnudge1980. We ran the coupled simulations

to evaluate how adding ocean and sea ice feedbacks impacted the signal from the optics change. Finally, we explored the long-term climate impacts of changing the optics by running a 39 year wind-nudged three member ensemble with prescribed sea ice and ocean. For this configuration, called F1850_UVnudge1980–2018, we nudged with ERA-I data from 1980–2018. We allowed the nudged winds to evolve over time in the F1850_UVnudge1980–2018 experiments to evaluate how interannual variability impacted the optics change.

We used several sets of temperature-dependent optics in our wind nudging experiments. For the experiment F1850_UVnudge1980, we ran the configuration with the control optics and with the temperature-dependent 240 K, 263 K, and 273 K optics. We continued with the 240 K and 273 K optics to evaluate the outer bounds of the effect from the temperature-dependent optics. We also added the 263 K optics set because that temperature was the closest to the average cloud temperature in the Arctic. For the B1850_UVnudge1980 and F1850_UVnudge1980–2018 experiments, we ran both of these configurations with the control

optics and the temperature-dependent 263 K optics.





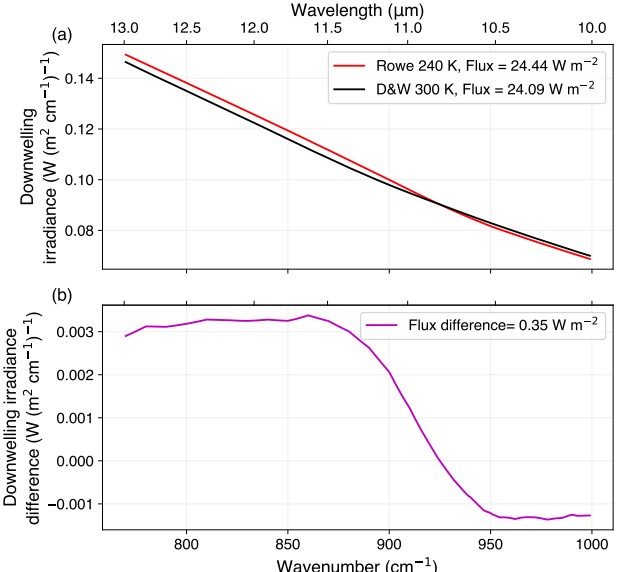

**Figure 5.** (a) Downwelling irradiance spectra of a 240 K supercooled liquid cloud ($\Delta z_{\mathrm{cld}} = 100$ m) modeled using a two-stream radiative transfer model between 770 and 1000 cm$^{-1}$. The spectra was modeled with both 240 K optics (red) and 300 K optics (black). (b) The difference between the spectra modeled with 240 K optics and 300 K optics.

## 3 Results

### 3.1 Two-stream radiative transfer model

We start with the influence of temperature-dependent optics on radiation in our simplest model, the two-stream radiative transfer model. As expected from Rowe et al. (2013), the downwelling irradiance and flux was higher for temperature-dependent optics than temperature-independent optics. The thinnest clouds (100 m thick with optical depth $\tau \sim 1$–1.5) showed the largest difference in downwelling flux between the temperature-dependent and temperature-independent optics (Fig. 5). For the 100 m thick cloud, all cloud temperatures had a 0.35 W m$^{-2}$ flux difference between the temperature-dependent and temperature-independent optics. However, as cloud thickness increased from 100 to 500 m ($\tau \sim 4$–8) and 1000 m ($\tau \sim 10$–15), the difference caused by our cloud optics change was negligible. We also want to note that the effect in our spectral model was half the size that Rowe et al. (2013) found for comparable surface and cloud temperatures (0.66 W m$^{-2}$), but our model was meant to be a proof of concept and not realistic, like Rowe et al. (2013)'s 16+ stream spectral model.

### 3.2 Single-Column Atmospheric Model Arctic Case Study

Next, we present results from the single-column atmospheric model runs for the Arctic field campaign MPACE held during October 2004. During almost the entire 17 day period, both temperature-independent and temperature-dependent optics produced



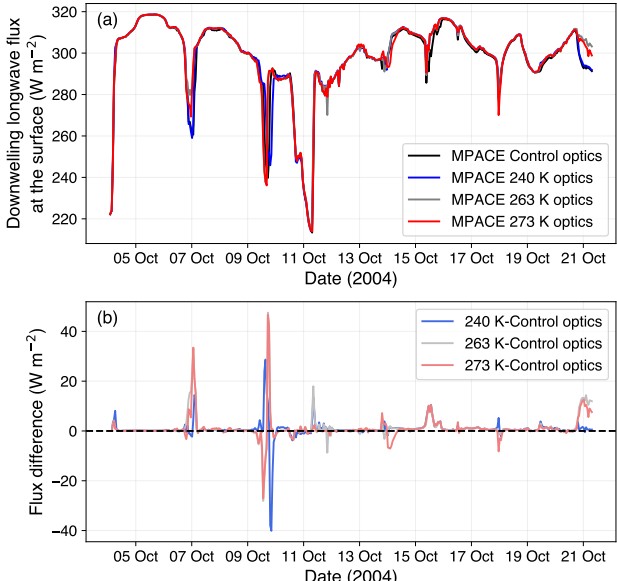

**Figure 6.** (a) Downwelling longwave flux at the surface modeled by SCAM for the MPACE IOP with four different sets of optics: Control - 298 K (black), 240 K (blue), 263 K (gray), and 273 K (red). (b) The difference in flux between the control and all three sets of temperature-dependent optics.

the same downwelling longwave flux at the surface (Fig. 6). The only notable differences (over 10 W m$^{-2}$) in downwelling longwave flux between the temperature-dependent optics and the control (temperature-independent) optics simulations occurred on the dates of 7, 10, 11, and 21 October 2004. On these dates, there were differences in cloud fraction and dominant cloud phase between the temperature-independent and temperature-dependent SCAM runs. Depending on these cloud type and amount disparities, flux differences were not consistently in one direction for all optics sets on a given date nor for one 180 optics set over the entire model run. In summary, cloud phase disparities between SCAM runs complicated the attribution of differences in flux to changes in the cloud optics.

Comparing downwelling longwave flux from temperature-independent and temperature-dependent optics amongst all cloud types yielded unclear results. Therefore, we focused our analysis of the cloud optics change on the cloud type where we anticipated the largest effect: supercooled liquid clouds. We isolated the impacts of the temperature-dependent optics by subsetting 185 the downwelling longwave flux, only including data points when there were low-level supercooled liquid clouds and the atmosphere was optically thin ($\tau < 5$). Table 2 described the results of this sub-setting. Notably, the medians of all the subsetted temperature-dependent fluxes were larger than the subsetted temperature-independent fluxes by 0.21 to 0.48 W m$^{-2}$. Although these flux differences were not large, they were similar to the flux differences we found in the two-stream radiative transfer model (Fig. 5). This result showed that the downwelling flux modeled by SCAM for low-level supercooled liquid clouds was 190 larger for the temperature-dependent optics. However, the differences between the temperature-dependent and temperature-independent flux medians were not statistically significant at the 95 % level (Table 2).



**Table 2.** Statistics from subsetted SCAM-MPACE downwelling longwave flux data

| Optics set | Median (W m$^{-2}$) | 95 % confidence interval on median (W m$^{-2}$) | Median$_{\text{optics set}}$-Median$_{\text{control optics}}$ (W m$^{-2}$) | Is the difference between medians statistically significant? |
| --- | --- | --- | --- | --- |
| Control | 307.09 | (305.39, 308.18) | – | – |
| 240 K | 307.37 | (306.19, 308.65) | 0.28 | No |
| 263 K | 307.57 | (306.32, 308.54) | 0.48 | No |
| 273 K | 307.28 | (305.84, 308.39) | 0.21 | No |

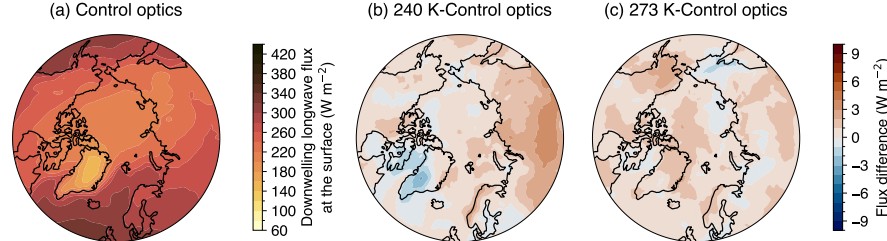

**Figure 7.** (a) The 40 year mean downwelling longwave flux at the surface from the F1850 control run. Flux differences in the 40 year averages between (b) the 240 K and control optics runs and (c) the 273 K and control optics runs from F1850. Stippling indicates the differences are statistically significant at a 95 % level following Wilks (2016). The colormaps are generated based on work by Crameri et al. (2020).

## 3.3 Freely evolving global climate model

Having shown that the temperature-dependent optics produce slightly more longwave flux than temperature-independent optics in both a simplistic radiative transfer model and a single-column atmospheric model, we next describe results from a freely
evolving atmosphere model run (F1850). For the F1850 experiment, the average downwelling longwave flux at the surface over the model run time span (40 years) was higher by $\sim$ 1–2 W m$^{-2}$ over most of the Arctic for the temperature-dependent optics than the temperature-independent optics (Fig. 7). However, these flux differences were not statistically significant at a 95 % level. The small magnitude of the flux differences was outweighed by the large variability in the annual mean flux, making the flux differences statistically insignificant. We also observed that the total area where temperature-dependent optics produced
more downwelling longwave flux was larger than where the temperature-independent optics produced more, but the spatial pattern of these areas was not consistent between Fig. 7b and Fig. 7c. This spatial inconsistency suggested, like the SCAM runs, that differences in the clouds between all three of the F1850 runs complicated our assessment of the impact on the flux from the temperature-dependent optics alone.





**Figure 8.** (a) The 1 year ensemble mean downwelling longwave flux at the surface from the F1850_UVnudge1980 control run. Flux differences in the 1 year ensemble averages between (b) the 240 K and control optics runs, (c) the 263 K and control optics runs, and (d) the 273 K and control optics runs from F1850_UVnudge1980. Stippling indicates the differences are statistically significant at a 95 % level following Wilks (2016). The colormaps are generated based on work by Crameri et al. (2020).

## 3.4 Wind-nudged global climate model

We next evaluate the impact of temperature-dependent optics on a wind-nudged atmosphere with an ensemble from the experiment F1850_UVnudge1980. The ensemble mean downwelling longwave flux at the surface from F1850_UVnudge1980 was higher ($\sim$ 1–7 W m$^{-2}$) in most of the Arctic for the temperature-dependent optics (Fig. 8). Critically, many flux differences were statistically significant, which showed that the temperature-dependent optics impacted longwave flux substantially in this modeling experiment. The flux differences became statistically significant in this experiment because the wind nudging re-

duced the variability in the annual mean flux between the ensemble members. Additionally, the region of the Arctic where the temperature-dependent optics produced more downwelling longwave flux was much larger for the F1850_UVnudge1980 experiments and more consistent between the temperature-dependent optics sets than the F1850 experiments. The spatial patterns of statistically significant flux differences for the F1850_UVnudge1980 experiments were also mostly consistent between the temperature-dependent optics sets. This high level of spatial consistency demonstrated that our results were not appreciably

affected by atmospheric circulation differences between the model runs due to the wind nudging.

    To understand the influence of ocean and sea ice coupling, we next describe the results from the coupled and dynamically constrained model ensemble (B1850_UVnudge1980). The ensemble mean of the downwelling longwave flux at the surface was higher ($\sim$ 1–3 W m$^{-2}$) in some of the Arctic for the temperature-dependent optics (Fig. 9). Surprisingly, no flux differences due to the temperature-dependent optics were statistically significant. Although both B1850_UVnudge1980 and

F1850_UVnudge1980 were nudged with the same winds, the nudging did not affect the variability in annual mean flux between ensemble members enough to make the flux differences significant. These results demonstrated that enabling coupling to the ocean and sea ice model components reduced the effect of the temperature-dependent optics in spite of the wind nudging.

    Finally, we detail the influence of temperature-dependent optics on a decadal time scale with a constrained atmosphere from our F1850_UVnudge1980–2018 experiment. The ensemble mean of the downwelling longwave flux at the surface was

higher ($\sim$ 1–2 W m$^{-2}$) in most of the Arctic for the temperature-dependent optics (Fig. 10). Some of these flux differences were statistically significant, which showed that the temperature-dependent optics impacted longwave flux substantially on a



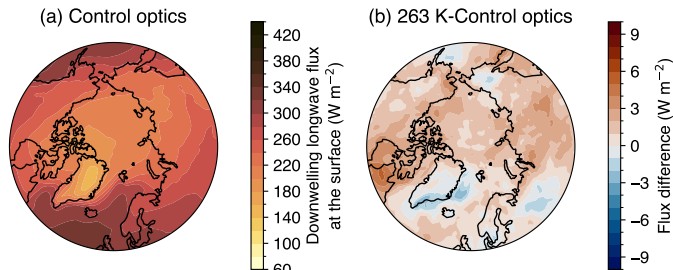

**Figure 9.** (a) The 1 year ensemble mean downwelling longwave flux at the surface from the B1850_UVnudge1980 control run. Flux differences in the 1 year ensemble averages between (b) the 263 K and control optics runs from B1850_UVnudge1980. Stippling indicates the differences are statistically significant at a 95% level following Wilks (2016). The colormaps are generated based on work by Crameri et al. (2020).

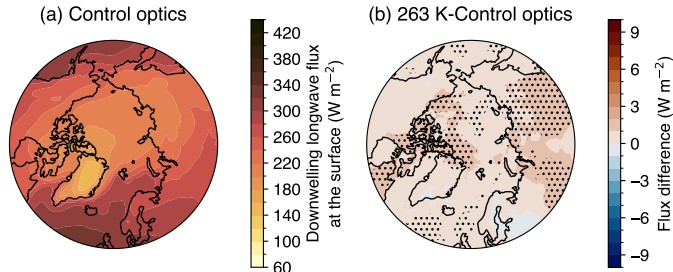

**Figure 10.** (a) The 39 year ensemble mean downwelling longwave flux at the surface from the F1850_UVnudge1980–2018 control run. Flux differences in the 39 year ensemble averages between (b) the 263 K and control optics runs from F1850_UVnudge1980–2018. Stippling indicates the differences are statistically significant at a 95 % level following Wilks (2016). The colormaps are generated based on work by Crameri et al. (2020).

decadal scale. However, the area and magnitude of statistically significant flux differences in the F1850_UVnudge1980–2018 ensemble (Fig. 10b) were smaller than the F1850_UVnudge1980 ensemble (Fig. 8c). This decades-long ensemble had fewer ensemble members, and thus a smaller sample size of the model climate's internal variability, and added interannual variability.
In summary, the effect of the temperature-dependent optics was widespread across the Arctic and statistically significant in some places, but the magnitude of the effect on a decadal time was only on the order of a few W m$^{-2}$.

Yet, this substantial impact on the longwave flux at the surface for the decades-long ensemble did not translate into an effect on surface temperature. Figure 11 shows Arctic near surface temperature anomalies for the F1850_UVnudge1980–2018 ensembles and the ERA-I data. The temperature trends for the control (-0.008 to 0.030 K decade$^{-1}$) and 263 K (-0.0019 to
-0.004 K decade$^{-1}$) F1850_UVnudge1980–2018 ensembles remained near zero and were minuscule compared to the ERA-I trend (0.650 K decade$^{-1}$).





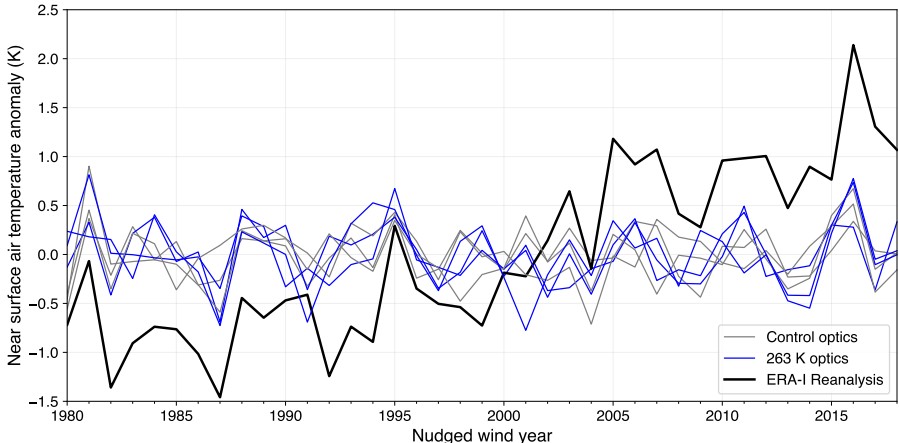

**Figure 11.** Near surface annual temperature anomalies from the F1850_UVnudge1980–2018 control optics (gray) and 263 K optics (blue)
ensembles and from the ERA-I reanalysis (black). The temperature anomalies were averaged over 60–90° N.

## 4  Discussion

Our main conclusion is that temperature-dependent optics are not a first priority for climate model radiation parameterization
development. While the optics did have a substantial impact on the mean state Arctic longwave radiation at range of model
temporal and spatial scales, their effect in the model experiment closest to the real world, the coupled ensemble, was minimal
and statistically insignificant. We found that the effect of the temperature-dependent optics was about 1–3 W m$^{-2}$, which
confirmed the results from Rowe et al. (2013) case study. Our novel model hierarchy worked, taking new physics and case
study results and finding a similar size effect on the climate. However, an effect of this magnitude has different implications
when considering a case study versus a global climate model. In the case study results from Rowe et al. (2013, 2022), they
concluded that the 1–2 W m$^{-2}$ effect of these optics mattered when retrieving cloud properties from radiance measurements
because retrievals of ice and liquid effective radii, ice fraction, and liquid water path were affected substantially. Whereas for
the global climate model, an effect of a few W m$^{-2}$ is within climate variability and thus relatively small. Additionally, the
optics didn't affect surface temperature trends in the decade-long wind-nudged ensemble. However, the effect of these optics
was not negligible and we recommend that model development add these optics to the list of parameterizations to be added
RRTMG.

This study has additional value in showing how a model hierarchy can be used to assess the importance of a model physics
change. In the first step, the two-stream radiative transfer model showed us that the optics change had an effect in a mathemati-
cal model (Fig. 12, panel 1), grounding evidence of our effect in the principles of radiative transfer. Second, the single-column
model showed us the effect in a fully parameterized atmospheric model at a single location (Fig. 12, panel 2). We found it
harder to isolate the effect of the optics in this model and it was at this level of model complexity we began to suspect that
the physics change might not produce a substantial effect. We also realized that internal variability and dynamical differences





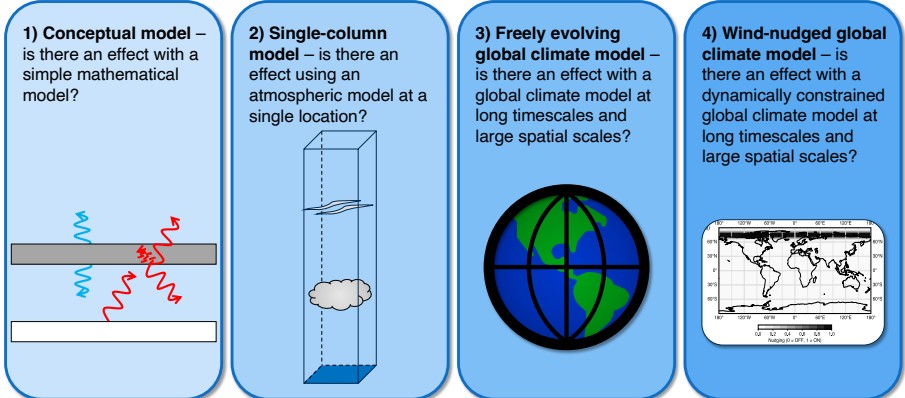

**Figure 12.** Process for detecting and evaluating the significance of a physics change in a model hierarchy of a conceptual model, single-column model, climate model, and wind-nudged climate model.

related to clouds between model runs might affect our results. The wind nudging step in the hierarchy allowed us to constrain the dynamic variability and amplify the signal in the radiation from the optics change. In the third step, we evaluated the optics change in an freely evolving global climate and found a statistically insignificant impact (Fig. 12, panel 3). The results at this

step confirmed our earlier suspicions that dynamics would obscure our ability to isolate the effect of the optics. The novel addition of wind nudging in the final step of this hierarchy allowed us to constrain the dynamic variability and amplify the signal in the radiation from the optics change to a statistically significant level (Fig. 12, panel 4). This hierarchy taught us a lot about the impact of our physics change because it told us at what model complexity and time and spatial scales the optics had an effect, as summarized in Table 3. As a result, this novel model hierarchy enabled us to make specific conclusions about the

effect of the optics and recommendations to the model development community.

There are some limitations and caveats of our study that we want to address. First, it is important to note that large uncertainties remain in the temperature-dependent optics in climatologically important spectral regions, including below 500 $\text{cm}^{-1}$ and from 1075 to 1575 $\text{cm}^{-1}$, where the temperature dependence is unknown (Rowe et al., 2020). Second, CESM2 CAM6 is known to have optically thick mean state Arctic clouds (McIlhattan et al., 2020) and we know from the case study results

(Rowe et al., 2013) that the effect of the temperature-dependent optics is greatest for clouds with liquid water paths of 1 to 10 $\text{g m}^{-2}$, which are optically thin. From this perspective, our study may be underestimating the effect of these optics. To address this, we could re-run steps three and four of our model hierarchy in a model with mean state optically thinner clouds, such as CESM2 CAM5 (McIlhattan et al., 2017). Yet, models like CAM5 with these thinner clouds underestimate supercooled liquid and overestimate ice in Arctic clouds (Kay et al., 2016). Considering these competing biases and the fact that CAM6's

supercooled liquid is more realistic than CAM5's (Gettelman et al., 2020; McIlhattan et al., 2020), we justify our choice to use CESM2 CAM6 in our model hierarchy. However, our model evaluation was based on performance in the Arctic and with respect to supercooled liquid. Scientists utilizing this hierarchy outside of the Arctic need to consider biases appropriate for their spatial domain and variables when choosing the global climate model for hierarchy steps three and four. Third, our conclusions



**Table 3.** Summary of the effect of temperature-dependent optics over the entire model hierarchy

| Experiment name | Model complexity | Spatial scale | Time scale | Ensemble members | Effect of optics | Substantial? |
|---|---|---|---|---|---|---|
| Two-stream radiative transfer model | Simple mathematical model | – | – | – | $0.35$ W m$^{-2}$ | No |
| SCAM | Full atmospheric model | One grid cell | Days (17) | – | $0.21$–$0.48$ W m$^{-2}$ | No |
| F1850 | Global climate model with prescribed ocean and sea ice | Entire Arctic (50–90° N) | Decades (4) | – | $1$–$3$ W m$^{-2}$ | No |
| F1850_UVnudge1980 | Global climate model with prescribed ocean and sea ice and nudged winds | Entire Arctic (50–90° N) | Year (1) | 10 | $1$–$7$ W m$^{-2}$ | Yes |
| B1850_UVnudge1980 | Fully coupled global climate model with nudged winds | Entire Arctic (50–90° N) | Year (1) | 10 | $1$–$3$ W m$^{-2}$ | No |
| F1850_UVnudge1980–2018 | Global climate model with prescribed ocean and sea ice and nudged winds | Entire Arctic (50–90° N) | Decades (4) | 3 | $1$–$2$ W m$^{-2}$ | Yes |

about the impact of temperature-dependent optics are limited to the models we used. We recommend using our model hierarchy

structure to test the effect of these optics in other climate models. For high resolution spectral radiation models, we recommend consulting Rowe et al. (2013) because they use a comparable model. Finally, the computational cost of fully implementing the temperature-dependent optics would be immense. In our study, we switched out the liquid optics lookup table, which didn't change the computational cost. Ideally, the model would match the cloud temperature and optics temperature by interpolating the optics properties. This implementation would mean the model performing that interpolation at every timestep and grid cell,

increasing the cost of the already costly radiation scheme significantly. One possible compromise to these two implementation approaches would be be to find the optics set closest to the cloud temperature and use that lookup table. We expect this third approach would be easy to implement and nominally increase the radiation scheme's computational cost.

Based on our study results, we have some suggestions for future work regarding both the temperature-dependent optics and the model hierarchy. First, the Antarctic and Southern Ocean have a high occurrence of supercooled liquid (Gettelman

et al., 2020), including optically thin supercooled liquid clouds at 240 K (Rowe et al., 2022). In addition, the atmosphere of the Antarctic interior is colder and drier than the Arctic, and there is evidence that liquid effective radii are smaller in the Arctic (Lubin et al., 2020), which would cause the temperature-dependent optics to have a larger effect (Rowe et al., 2013).



These factors make the Antarctic a prime second location to test the effect of the temperature-dependent optics, specifically the wind-nudging experiments from the model hierarchy. Furthermore, the effect of the temperature-dependent optics is also of

import for upwelling infrared radiation, which is expected to have a magnitude about twice as large in the tropics as the effect on downwelling infrared radiation in the Arctic for a supercooled liquid cloud at 240 K and typical atmospheres (Rowe et al., 2013). Another avenue for future research is using our novel model hierarchy to evaluate the impact of other potential model physics additions. For example, Meng et al. (2022) developed a new dust particle size distribution for CESM that improved the representation of super coarse dust. However, their work didn't assess the new dust parameterization outside of the dust

size distribution and our model hierarchy could be used to evaluate this parameterization's impact on cloud properties, aerosol optical depth, aerosol radiative forcing, etc. before it is incorporated into CESM. Taking a step back from individual parameterizations, this model hierarchy could even be used to detect changes between different versions of radiation or microphysical schemes.

## 5 Conclusions

In this study, we assessed the impact of temperature-dependent liquid water optics on longwave radiation in Arctic over a hierarchy of models. Our model hierarchy, increasing in complexity, included a mathematical two-stream radiative transfer model, a single-column atmospheric model, a freely evolving global climate model, and a wind-nudged global climate model. We found that the optics had insubstantial effects on the order of $0.1$ W m$^{-2}$ for both the two-stream and single-column models. For the freely evolving global climate model (CESM2), the optics had a $1$–$3$ W m$^{-2}$ effect on a decadal time scale

that we deemed insubstantial because of high interannual variability within the model. In the wind-nudged model ensemble at a year-long time scale, the optics had a substantial $1$–$7$ W m$^{-2}$ effect in an atmosphere-only configuration and an insubstantial $1$–$3$ W m$^{-2}$ effect in a coupled configuration. This result demonstrated that constraining the dynamic variability through wind-nudging amplified the non-dynamical signal of the temperature-dependent optics, but adding coupled ocean and sea ice components to the model and making it more realistic, reduced the impact of the optics significantly. Finally, with a wind-

nudged atmosphere-only ensemble on a decadal scale, we found that the optics had a substantial $1$–$2$ W m$^{-2}$ effect in the Arctic. Our first conclusion is that given the magnitude of the optics' effect on longwave radiation at various model, time, and spatial scales, the temperature-dependent optics should eventually be added to radiation parameterizations, but that they are not a first priority. Our second conclusion is that the model hierarchy we developed can be used to assess the importance of model physics changes, such as new parameterizations or entire schemes.

## Appendix A: Equations

### A1 Two-stream radiative transfer model

We used a gamma distribution to represent the droplet size distribution as a function of the droplet radius ($n(r)$). The distribution of radii covered 0.01 to 50 μm with an increment of 1 μm. The following equations described the properties of the gamma



distribution as outlined in Petty (2006a):

$$\alpha = 3, \tag{A1}$$

$$n(r) = ar^\alpha e^{-br}, \tag{A2}$$

$$b = \frac{\alpha + 3}{r_{\mathrm{eff}}}, \tag{A3}$$

$$a = \frac{Nb^{\alpha+1}}{\alpha!}, \tag{A4}$$

where $\alpha$, $a$, and $b$ were the parameters that describe the gamma distribution.

The extinction efficiency ($Q_{\mathrm{ext}}$) and scattering efficiency ($Q_{\mathrm{sca}}$) were calculated as functions of wavenumber ($\tilde{\nu}$) and droplet radius ($r$) using the method in Wiscombe (1979). We used the wavenumbers 770 to 1000 $\mathrm{cm}^{-1}$ with an increment of 1 $\mathrm{cm}^{-1}$. We calculated the volume extinction coefficient ($\beta_{\mathrm{ext}}$) and the single-scattering albedo ($\tilde{\omega}$) solely as functions of wavenumber with the following equations from Petty (2006a):

$$\beta_{\mathrm{ext}} = \int_0^r n(r) Q_{\mathrm{ext}}(r) \pi r^2 \, \mathrm{d}r, \tag{A5}$$

$$\tilde{\omega} = \frac{1}{N} \int_0^r n(r) \frac{Q_{\mathrm{sca}}(r)}{Q_{\mathrm{ext}}(r)} \, \mathrm{d}r. \tag{A6}$$

Then, we took $\beta_{\mathrm{ext}}$ and $\tilde{\omega}$ and calculated the following quantities for the two-stream radiative transfer model as functions of wavenumber from Petty (2006b):

$$\tau^* = \beta_{\mathrm{ext}} \Delta z_{\mathrm{cld}}, \tag{A7}$$

$$\Gamma = 2\sqrt{1 - \tilde{\omega}}\sqrt{1 - g\tilde{\omega}}, \tag{A8}$$

$$r_\infty = \frac{\sqrt{1 - g\tilde{\omega}} - \sqrt{1 - \tilde{\omega}}}{\sqrt{1 - g\tilde{\omega}} - \sqrt{1 - \tilde{\omega}}}, \tag{A9}$$

$$r = \frac{r_\infty[e^{\Gamma\tau^*} - e^{-\Gamma\tau^*}]}{e^{\Gamma\tau^*} - r_\infty^2 e^{-\Gamma\tau^*}}, \tag{A10}$$

$$t = \frac{1 - r_\infty^2}{e^{\Gamma\tau^*} - r_\infty^2 e^{-\Gamma\tau^*}}, \tag{A11}$$

where $\Delta z_{\mathrm{cld}}$ was the the cloud thickness, $\tau^*$ was the cloud optical depth, $\Gamma$ was a parameter, $r_\infty$ was the albedo of a semi-infinite cloud, and $r$ and $t$ were the cloud reflectance and transmittance over a black surface, respectively. Since we assumed in our two-stream model used a snow surface, we calculated the following quantities as functions of wavenumber to find the





cloud properties over a non-black surface from Petty (2006b):

$$\tilde{r} = \frac{r + r_{\text{sfc}}t^2}{1 - r_{\text{sfc}}r}, \tag{A12}$$

$$\tilde{t} = \frac{t}{1 - r_{\text{sfc}}r}, \tag{A13}$$

$\quad \varepsilon = 1 - \tilde{r} - \tilde{t}, \tag{A14}$

where $\tilde{r}$, $\tilde{t}$, and $\varepsilon$ were the cloud reflectance, transmittance, and emissivity over a non-black surface, respectively, and $r_{\text{sfc}}$ was the surface albedo. We also assumed that emissivity was equal to absorptivity according to Kirchhoff's Law. Finally we calculated the downwelling longwave spectra ($F_{\downarrow,\tilde{\nu}}$) as a function of wavenumber and the flux ($F_{\downarrow}$) for the cloud at the surface with the following equations from Petty (2006b):

$\quad F_{\downarrow,\tilde{\nu}} = \tilde{r}_{\tilde{\nu}} B_{\tilde{\nu}}(T_{\text{sfc}})(1 - r_{sfc}) + \varepsilon_{\tilde{\nu}} B_{\tilde{\nu}}(T_{\text{cld}}), \tag{A15}$

$$F_{\downarrow}(\tilde{\nu}_1, \tilde{\nu}_2) = \int\limits_{\tilde{\nu}_1}^{\tilde{\nu}_2} F_{\downarrow,\tilde{\nu}}\, d\tilde{\nu}, \tag{A16}$$

where $B_{\tilde{\nu}}(T)$ was the Planck blackbody function and $T_{\text{sfc}}$ and $T_{\text{cld}}$ were the temperatures of the surface and cloud, respectively.

## A2 CESM2 optics calculation

The CESM2 radiation scheme has 16 longwave bands and 14 shortwave bands, as shown in Tables A1 and A2, respectively.
The optics file contains the following variables for both longwave and shortwave bands: mass extinction coefficient ($k_{\text{ext}}$), mass absorption coefficient ($k_{\text{abs}}$), mass scattering coefficient ($k_{\text{sca}}$), single-scattering albedo ($\tilde{\omega}$), asymmetry parameter ($g$), extinction efficiency ($Q_{\text{ext}}$), absorption efficiency ($Q_{\text{abs}}$), and scattering efficiency ($Q_{\text{sca}}$). Each variable has the dimensions $\mu$ and $\lambda$. These parameters describe the gamma distribution that defined the droplet size distribution ($n(D)$) as a function of droplet diameter in the following equations:

$\quad 2 < \mu < 15, \tag{A17}$

$$\frac{\mu + 1}{50 \times 10^{-6}\ \text{m}} < \lambda < \frac{\mu + 1}{2 \times 10^{-6}\ \text{m}}, \tag{A18}$$

$$n(D) = \frac{\lambda^{\mu+1}}{\Gamma(\mu+1)} D^{\mu} e^{-\lambda D}, \tag{A19}$$

$$N = \frac{\Gamma(\mu+1)}{\lambda^{\mu+1}}, \tag{A20}$$

$$D_{\text{eff}} = \frac{\mu + 3}{\lambda}, \tag{A21}$$

$\quad x = \frac{\pi D_{\text{eff}}}{\lambda}, \tag{A22}$

where $N$ was the droplet number concentration, $D_{\text{eff}}$ was the effective droplet diameter, and $x$ was the size parameter. For equation A22, $\lambda$ represented wavelength, but for equations A17-A21, $\lambda$ was the droplet size distribution parameter.





**Table A1.** RRTMG longwave bands

| Band index | Band minimum (µm) | Band maximum (µm) | Band midpoint (µm) | Band minimum ($\text{cm}^{-1}$) | Band maximum ($\text{cm}^{-1}$) | Band midpoint ($\text{cm}^{-1}$) |
|---|---|---|---|---|---|---|
| 1 | 28.57 | 1000.0 | 169.03 | 10 | 350 | 59 |
| 2 | 20.00 | 28.57 | 23.90 | 350 | 500 | 418 |
| 3 | 15.87 | 20.0 | 17.82 | 500 | 630 | 561 |
| 4 | 14.29 | 15.87 | 15.06 | 630 | 700 | 664 |
| 5 | 12.20 | 14.29 | 13.20 | 700 | 820 | 758 |
| 6 | 10.20 | 12.20 | 11.16 | 820 | 980 | 896 |
| 7 | 9.26 | 10.20 | 9.72 | 980 | 1080 | 1029 |
| 8 | 8.47 | 9.26 | 8.86 | 1080 | 1180 | 1129 |
| 9 | 7.19 | 8.47 | 7.81 | 1180 | 1390 | 1281 |
| 10 | 6.76 | 7.19 | 6.97 | 1390 | 1480 | 1434 |
| 11 | 5.56 | 6.76 | 6.13 | 1480 | 1800 | 1632 |
| 12 | 4.81 | 5.56 | 5.17 | 1800 | 2080 | 1935 |
| 13 | 4.44 | 4.81 | 4.62 | 2080 | 2250 | 2163 |
| 14 | 4.20 | 4.44 | 4.31 | 2250 | 2380 | 2319 |
| 15 | 3.85 | 4.20 | 4.01 | 2380 | 2600 | 2493 |
| 16 | 3.08 | 3.85 | 3.44 | 2600 | 3250 | 2907 |

We calculated $Q_{\text{ext}}$, $Q_{\text{sca}}$, and $g$ as functions of wavenumber with the method outlined in Wiscombe (1979). For the rest of the variables in the file, we used the following equations:

$$Q_{\text{abs}} = Q_{\text{ext}} - Q_{\text{sca}}, \tag{A23}$$

$$\tilde{\omega} = \frac{Q_{\text{sca}}}{Q_{\text{ext}}}, \tag{A24}$$

$$k_{\text{abs}} = \frac{3 Q_{\text{abs}} \lambda}{2 \rho_w (\mu + 3)}, \tag{A25}$$

$$k_{\text{ext}} = \frac{3 Q_{\text{ext}} \lambda}{2 \rho_w (\mu + 3)}, \tag{A26}$$

$$k_{\text{sca}} = \frac{3 Q_{\text{sca}} \lambda}{2 \rho_w (\mu + 3)}, \tag{A27}$$

where $\lambda$ and $\mu$ were the parameters for the droplet size distribution and $\rho_w$ was the density of water. For each longwave and shortwave band, we calculated each variable at the band maximum, midpoint, and minimum. Then, we took the average of those three values and saved that average value to the optics file for that band.



**Table A2.** RRTMG shortwave bands

| Band index | Band minimum (μm) | Band maximum (μm) | Band midpoint (μm) | Band minimum (cm$^{-1}$) | Band maximum (cm$^{-1}$) | Band midpoint (cm$^{-1}$) |
|---|---|---|---|---|---|---|
| 1 | 3.077 | 3.846 | 3.440 | 2600 | 3250 | 2907 |
| 2 | 2.500 | 3.077 | 2.773 | 3250 | 4000 | 3606 |
| 3 | 2.150 | 2.500 | 2.319 | 4000 | 4650 | 4313 |
| 4 | 1.942 | 2.150 | 2.043 | 4650 | 5150 | 4894 |
| 5 | 1.626 | 1.942 | 1.777 | 5150 | 6150 | 5628 |
| 6 | 1.299 | 1.626 | 1.453 | 6150 | 7700 | 6881 |
| 7 | 1.242 | 1.299 | 1.270 | 7700 | 8050 | 7873 |
| 8 | 0.778 | 1.242 | 0.983 | 8050 | 12850 | 10171 |
| 9 | 0.625 | 0.778 | 0.697 | 12850 | 16000 | 14339 |
| 10 | 0.442 | 0.625 | 0.525 | 16000 | 22650 | 19037 |
| 11 | 0.345 | 0.442 | 0.390 | 22650 | 29000 | 25629 |
| 12 | 0.263 | 0.345 | 0.301 | 29000 | 38000 | 33196 |
| 13 | 0.200 | 0.263 | 0.229 | 38000 | 50000 | 43589 |
| 14 | 3.846 | 12.195 | 6.849 | 820 | 2600 | 1460 |

*Code and data availability.* The temperature-dependent CRI and optics and the processed SCAM and CESM2 data are available from
https:///doi.org/10.5281/zenodo.12587014 (Gilbert, 2024a). The code needed to run all of the model hierarchy experiments and the namelists
for the SCAM and CESM2 experiments are available from https:///doi.org/10.5281/zenodo.12612342 (Gilbert, 2024b).

*Author contributions.* AG and JEK designed the experiments and model hierarchy. AG performed the model runs and analysis. JEK and PR
provided input on the analysis. AG wrote the paper, with input from all authors.

*Competing interests.* The authors declare that they have no conflict of interest.

*Acknowledgements.* AG was funded by the National Science Foundation (NSF) Graduate Research Fellowship (grant no. 2040434), NSF
(grant no. 2233420), and the PREFIRE Mission (NASA grant no. 849K995). JEK was funded by PREFIRE and NSF (grant no. 2233420).
PR was funded by NSF (grant no. 2127632). Some Cheyenne core hours were provided by the Polar Climate Working Group at the National
Center for Atmospheric Research. All authors thank the Polar Climate Working Group for helpful feedback.



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
