# Peer review of "A Novel Model Hierarchy Isolates the Limited Effect of Supercooled Liquid Cloud Optics on Infrared Radiation"

_EGUsphere, 2024_

## Referee Comment (RC2)

**Review of "A Novel Model Hierarchy Isolates the Effect of Temperature-dependent Cloud Optics on Infrared Radiation" by Gilbert *et al.* submitted to *Geoscientific Model Development***

This study investigated the effect of "temperature-dependent cloud optics" on infrared radiation, with a specific focus on the Arctic region. The analysis is done through a combination of a simple mathematical model for two-stream radiative transfer, a single-column atmospheric model, an atmospheric model, and a wind-nudged atmospheric model. The results suggest that the impact of "temperature-dependent cloud optics" is less significant compared to the internal variability in the Arctic region. When model winds are nudged towards reanalysis, the internal variability is partially constrained, and the effect of temperature-dependent cloud optics becomes more prominent.

This study has the potential to update our understanding of the impact of temperature-dependent cloud optics on climate simulations. However, there are a few major issues in this manuscript which I list below. The authors may need to perform additional experiments and data analyses. And based on that, I would recommend major revision.

1. It is not correct to claim that the designed model simulations study the effect of "temperature-dependent cloud optics". The authors simply switched the cloud optics at 298 K in the original model to the cloud optics at other temperatures. It is essentially cloud optics at a constant temperature (or temperature-independent cloud optics). While it is OK to simply do this in idealized single-column model experiments, because the cloud temperature can be set at any value to quantify the flux changes in the extreme cases, it is not appropriate to do this in the full atmospheric model simulations. Although the authors mentioned in the discussion section that this will be part of future work, "temperature-dependent" is still a confusing term to describe the current approach. I recommend the authors rephrasing it or implementing the physics to the atmospheric model.

2. Based on what has been presented in this manuscript, I don't think the analyses are sufficiently thorough, and the power of model hierarchy on understanding the impact of physical assumptions in climate models is not fully realized in this study. For example, the change of surface downward longwave radiative flux due to the use of temperature-dependent cloud optics is not well quantified. Only spatial pattern of differences between model runs are shown (Figures 7~10). The ranges given in the manuscript are mostly approximate (e.g., 1~2 W/m², 1~3 W/m², 1~7 W/m², etc.). Also, only surface downward longwave flux changes are quantified here, but the impact on OLR is also important from the perspective of the TOA radiation budget. I suggest that the authors should start from analyzing the global mean and regional mean time series of OLR and surface downward longwave flux, providing an estimate of flux differences, and then go further to analyze the spatial pattern of flux changes.

3. For the two-stream radiative transfer model described in section 2.3, the authors chose to use a very simple mathematical model to do the calculation. This does not take into account the atmospheric absorption, while it is an important factor that may mask the effect of cloud optics change. The authors may use a more developed two-stream radiative transfer model. For example, RRTMG_LW provides a single-column version that users can specify any profile to

test. Using this model, the authors can calculate the flux differences in broad cases and even plot the sensitivity of flux difference to the meteorological factors and cloud properties.

4. For the single-column atmospheric model, what variables are prescribed by the observations? My understanding is that clouds are not constrained by the observations. For most observational period in Figure 6, the flux difference is very close to 0. Are they cloud-free scenarios? I would suggest filtering out the clear-sky cases and focus on the cloudy scene.

**Specific Comments**

1. L19-21: A reference may be necessary to support the statement that "All else being equal, clouds with small particle sizes also scatter more shortwave and emit more downwelling longwave than clouds with large particle sizes.

2. L39-40: "Specifically, temperature-dependent liquid water optics are not used in RRTMG." Related to the first major issue, this sentence is very confusing as the authors did not implement the full temperature-dependent liquid water optics in the model, either. The authors may be more specific on what specific cloud optics RRTMG has used (e.g., at 298 K), and point out that this may not reflect the truth in the supercooled liquid cloud regime.

3. L39: Also cite Clough et al. (2005; https://doi.org/10.1016/j.jqsrt.2004.05.058)

4. L45-47: This long sentence is a bit confusing. "supercooled liquid clouds frequently occur in both observations […] and the climate model […] and where the atmosphere is typically cold and dry." These three are not in parallel. Consider this alternative: "supercooled liquid clouds frequently occur in the cold and dry region, as evidenced by observations and climate model simulations."

5. L92: For surface, "albedo" is specific for solar radiation. A better term could be "reflectivity".

6. Figure 2: "reflected ground emission" is ambiguous. A better alternative is "ground emission scattered by clouds"

7. Figure 2: In longwave radiative transfer, it better aligns with the convention to use emissivity rather than reflectivity.

8. Figure 3: For panels (c) and (d), it could be better to visualize the difference between 263 K optics and CESM optics.

9. Table 1: Do these model runs include model spin-up period? It takes time for the model to adjust to the new state.

10. Table 1: Why is the 263 K run missing in the F1850 experiment? Especially consider that Figure 3 highlights the comparison between 263 K optics and CESM optics, and also the 263 K run appears in all other experiments.

11. L141: "the next time step". Note that 6-hourly ERA-Interim reanalysis is used here while the model step is 30 minutes by default. According to the referred literature, this is indeed the next available analysis time, not the next model time. Please be more specific and clear.

12. Figure 4: In panel (a), I noticed that there is a smoothing gradient at the boundaries of the latitudinal band. The previous study cited by the authors explicitly mentioned that they

applied smoothing (by setting $\alpha$ to a value between 0 and 1 in some region). Did the authors also apply the same technique? Also, in panel (b), a solid line is connected between $\alpha = 0$ and $\alpha = 1$ at around 800 hPa. Is the smoothing technique also applied here? To make it clear, instead of using line plot, the authors may choose scatter plot instead to visualize the exact $\alpha$ values at each discrete layer.

13. L164-165: "the downwelling irradiance and flux was higher for temperature-dependent optics than temperature-independent optics" This is confusing. It would be better to state that the downwelling irradiance and flux was higher for cloud optics at X temperature than the optics at Y temperature.

14. L165: "The thinnest clouds […] showed the largest difference." This statement is not supported by Figure 5, as no results are presented for clouds at different thickness.

15. L167: What is the meaning of "all cloud temperatures"? Rephrase this sentence.

16. L168-169: "However, as cloud thickness increased from 100 to 500 m […]" This is not shown in any figure.

17. L170~171: "but our model was meant to be a proof of concept and not realistic". Why not use a realistic model, given that a quantitative estimate of the effect is provided above (0.35 W/m$^2$)?

18. L177-179: The authors mentioned that when cloud optics at different temperatures are used, the cloud fraction and cloud phase in the simulations are different. I assume that the authors do not prescribe the model simulations with observed clouds. What are the differences in cloud fraction and properties exactly? Having these differences, I don't think this is an apple-to-appple comparison to show the net effect of cloud optics at different temperatures since cloud variability has played a role.

19. Figure 7: I don't see stippling in the figure, so it is better to say that no significance in the figure caption.

20. Figure 8: What's the regional mean difference in these plots? The average can be performed over 50ºN~90ºN, consistent with the given latitudinal band in Table 3, and the values can be added to the panel title.

21. L208: I suggest adding "at 5% significance level" to be more accurate and specific.

22. L209-210: "because the wind nudging reduced the variability in the annual mean flux between the ensemble members" A figure may be necessary to show this. If there are too many figures, consider combining the information in one figure. For instance, Figures 7~10 show similar information and can be merged into one figure.

23. L218~220: "no flux differences […] were statistically significant" Instead of setting some threshold, I suggest providing a $p$-value so that we can understand how far it is from the significance threshold.

24. L232-236: Given that the authors simply change the cloud optics at another temperature, the effect on mean 2-m air temperature difference should be more prominent than the effect on 2-m air temperature trend, since the temperature-cloud property feedback is muted. Also,

considering that no greenhouse gas and aerosol forcings are included in the simulations, it makes no sense to compare to the ERA-I 2-m air temperature trend.

25. L246-247: "Whereas for the global cliate model, an effect of a few W m$^{-2}$ is within climate variability and thus relatively small." Note that the historical change in effective radiative forcing from 1750 and 2019 is also a few W m$^{-2}$.

26. Table 3: The values in the "Effect of optics" column should be the regional mean values as defined in the "Spatial scale" column.

---

## Author Comment (AC1)

**Response to Reviewer #1 – Gilbert et al. (under review) GMD**

Reviewer comments are in black. Author responses are in blue. *Changes to manuscript are in*
*italic.*

We thank the reviewer for their time and constructive review. We provide a point-by-point
response below.

This manuscript seeks to understand how a change in the specification of the optical properties
of liquid clouds might affect the simulation of Arctic climate. The change is motivated on physical
grounds - the index of refraction of water is temperature dependent but this dependence is
usually ignored when mapping cloud physical to cloud optical properties in broadband codes.
Mie calculations are performed for drop size distributions consistent with the CESM2 climate
model using indexes of refraction valid at 273, 263, and 240K; these are used in an off-line
radiative transfer model across limited spectral ranges in the infrared, as well as in single-column
simulations, uncoupled and coupled freely-running ensembles, and in an ensemble in which
winds are nudged towards reanalysis at high latitudes. Small differences in spectrally-resolved
fluxes in the offline simulations; differences in integrated fluxes are lost in the variability in single-
column and free-running model simulations, becoming  large enough to be distinguished from
noise, if still small, in the nudged simulations.

The manuscript has two goals: 1) to assess the possible impact of an elaboration of cloud optics
on simulations in the Arctic, and 2) to develop methods for such an assessment. Both goals might
be reached more effectively by revisiting the experimental design to more clearly delineate the
perturbation that might be expected from such a change from any subsequent impact on
simulations.

The motivation for the study is well-grounded: the index of refraction of liquid water does indeed
depend on temperature, so the degree to which fluxes might be systematically biased in some
circumstances is not know a priori. What can be anticipated, however, is that the impact on
fluxes will be restricted to thin clouds, since (band-wise) fluxes will only change when (band-
wise) optical thickness is in the range of roughly 0.5 - 3 i.e. where the clouds are neither optically
thin or thick. (That this is not illustrated using the two-stream model is a missed opportunity.) The
impact of changes in the index of refraction thus depends on the population of clouds and
atmospheric states. The magnitude of this change for a given population of clouds, such as those
produced by a particular climate model, could be evaluated with off-line broadband radiation
calculations.

That this impact of the optics is limited to thin clouds is illustrated in the paper by the two-stream
radiative transfer model. From the unrevised paper L164-169:

As expected from Rowe et al. (2013), the downwelling irradiance and flux was higher for
temperature-dependent optics than temperature-independent optics. The thinnest clouds (100
m thick with optical depth $\tau \sim 1$–1.5) showed the largest difference in downwelling flux between the temperature-dependent and temperature-independent optics (Fig. 5). For the 100 m thick
cloud, all cloud temperatures had a 0.35 W m−2 flux difference between the temperature-
dependent and temperature-independent optics. However, as cloud thickness increased from
100 to 500 m ($\tau \sim$ 4–8) and 1000 m ($\tau \sim$ 10–15), the difference caused by our cloud optics
change was negligible.

We appreciate the reviewer's framing for the estimation the supercooled liquid optics effect and
have modified the paper to reflect it:

L43-46 revised paper:

*For instance, using a high spectral resolution line-by-line radiative transfer model applied to case*
*studies in the Arctic, Rowe et al. (2013) found that these supercooled liquid water optics can*
*increase modeled longwave fluxes emitted by thin (liquid water path < 10 g m$^{-2}$) supercooled*
*liquid-containing clouds by up to 1.7 W m$^{-2}$.*

L51-52 revised paper:

*We focus on the Arctic because it is a cold and dry region where thin supercooled liquid clouds*
*frequently occur in observations (Cesana et al., 2012) and climate model simulations (McIlhattan*
*et al., 2020).*

The title and framing of the manuscript is misleading: tests in the dynamical models do not use
temperature-dependent cloud optics; rather they replace cloud optics computed with the index
of refraction used at a single temperature with optics computed at a different temperature.
Whether accounting for the temperature dependence of cloud optical properties would impact
fluxes and/or other simulation characteristics can not be assessed with the current information.

We appreciate this important communication issue brought up by the reviewer, which was also
brought up by reviewer #2. We agree with the reviewer's comment that the authors are not using
"temperature-dependent" cloud optics, but optics at a single temperature applied to liquid water
at all temperatures. Therefore, we have changed the language "temperature-dependent" optics
in the paper to "supercooled liquid" optics to accurately reflect our methodology, results, and
conclusions.  We have also changed the language "temperature-independent" optics to "room
temperature" optics.

The authors assert that the computational cost of implementing cloud optics that depend on
temperature would be "immense" (lines 282-287) but this is unlikely to be true: cloud optics are
usually a tiny portion of the time spent in radiation calculations.

We agree that using the word "immense" is unnecessary and have removed it, but fully
implementing the temperature-dependent optics would add time & complexity. In climate model
development, it is best practice to not add time and complexity to the model unless necessary.

Considering the modest impacts these temperature-dependent optics would on climate,
implementing them is not a first priority.

Fully implementing the temperature-dependent optics would involve taking the grid box
temperature, which would presumably fall between two sets of supercooled liquid optics, and
linearly interpolating the two sets of optics the temperature fell between to create a set of liquid
water optics to perfectly match the grid box temperature. This process would have to be repeated
for each grid box in the atmosphere containing a cloud, at every time step the radiation code is
run. This implementation would not add immense time and complexity to the radiation code, but
it wouldn't be negligible either. This cost is why we recommend the implementation described
L285-287 of the unrevised paper. It involves matching the grid box temperature to the
temperature of the closest supercooled liquid optics set available and using that optics set. This
implementation does add time and complexity, but considerably less than the first option.

L251-258 revised paper:

*Finally, fully implementing the supercooled liquid water optics would increase the model*
*computational cost. In our study, we switched out the liquid optics lookup table, which didn't*
*change the computational cost. Ideally, the model would match the cloud temperature and*
*optics temperature by interpolating the optics properties. This implementation would involve the*
*model performing that interpolation at every timestep and grid cell, increasing the cost of the*
*already costly radiation scheme significantly. One possible compromise to these two*
*implementation approaches would be to find the optics set closest to the cloud temperature and*
*use that lookup table. We expect this third approach would be easy to implement and nominally*
*increase the radiation scheme's computational cost.*

*Interpretation*

The motivation for the "model hierarchy" is not made clear. The answers to the questions on lines
74-83 are tautological, i.e. the single column model is motivated by asking what the impact is at a
given location during a finite time frame. Linking each set of simulations to a testable hypothesis
will help readers make sense of results.

We agree with the reviewer and the motivation and clarity of the model hierarchy was also
brought up by reviewer #1. In response to both reviewer comments, the authors have decided to
restructure the paper and the model hierarchy as a function of dynamical constraint, not model
complexity. As a part of the restructuring, we removed the questions in the model hierarchy
outline, as they no longer felt appropriate to the authors.

Parametric sensitivity studies, as in Rowe et al. 2013 and as might be done with the spectrally-
resolved model are useful in motivating the work. A missing step is broadband calculations
analogous to those used in the global model simulations - say, offline calculations with RRTMG
over the distribution of Arctic clouds produced by CESM - to understand how those parametric
sensitivities convolve with the population of Arctic clouds in the model to be examined and whether one might expect systemic differences in interactive simulations. It is unclear what is
gained from the wind-nudged simulations, which are motivated by trying to constrain internal
variability, that wouldn't emerge more clearly from calculations applying changed optics to the
clouds produced by the unperturbed model.

We are concerned that the reviewer thinks that it is "unclear what is gained from the wind-nudged
simulations". Our results show that the only simulations in which the optics change is detected
are when the wind-nudging has been applied. In the "unperturbed" model (i.e., the freely evolving
global model), the signal from the optics change is hard to detect and small compared to the
model generated variability. We explain more here for the reviewer to address this confusion.

In the freely evolving global climate model, simulations with have different sequencing of
atmospheric events. These differing sequences of atmospheric events can make it hard to detect
a difference due to a cloud optics change. For example, the control optics simulation might have
produced an extra-tropical storm in the Arctic in August of year 15 whereas the 240 K optics
simulations might not have due to differing atmospheric sequencing due to inherent chaos (i.e.,
atmospheric internal variability). So, when we compare the mean downwelling longwave at the
surface between the control optics and supercooled liquid optics simulations, there are different
sequences of atmospheric events in each mean downwelling longwave timeseries in addition to
any difference caused by the optics alone.

The wind nudging is an attempt to remove the "noise" of different atmospheric sequences due to
the chaos from the signal of the supercooled liquid optics. By dynamically constraining both the
control and supercooled liquid optics simulations to the same atmospheric sequence, both
simulations produce the same sequence of the winds and large atmospheric circulation.
Therefore, the noise caused by different sequences of storms and clouds is reduced and the
signal from the supercooled liquid optics is emphasized. In short, the wind nudging increases the
signal-to-noise ratio for the longwave effect of supercooled liquid optics.

We believe this confusion has arisen from our text in the original manuscript not being clear.
Thus, we have also modified the text to make these points clearer as well at multiple points in the
revised manuscript.

L30-31 revised paper:

*A key advantage of prescribing the winds using nudging is that the time evolution of the prescribed*
*and modeled large-scale circulation is synchronized to the prescribed wind time evolution.*

L36-38 revised paper:

*These studies show that wind nudging is a powerful tool to amplify a radiative signal above*
*chaotic atmospheric noise by constraining the time sequence of the modeled atmospheric*
*circulation.*

*We anticipate using this hierarchy of constraint on the modeled atmospheric circulation sequencing will be of value. We expect the most dynamically constrained models will enable the easiest detection of the optics change. In contrast, dynamically unconstrained models will have more noise from internal climate variability and that noise may make it hard to detect the optics change signal.*

*Nudging the winds constrains the internal variability of the modeled climate system to a specific sequence of atmospheric circulation, which was the ERA-I winds in our experiments. Since all experiments were constrained to the same atmospheric circulation sequence, they were all likely to model the same sequence of clouds.*

Single-column model simulations can be expected to diverge somewhat in response to even tiny changes, making the interpretation of changes on particular days in a long simulation ambiguous. (Did the authors consider doing ensembles of single-column model simulations to see if the cloud optics change can be teased out?).

We thank the reviewer for raising this issue. The single-column model we used was relaxed to temperature and aerosol observations and the dynamics were prescribed. Therefore, given the constraints on the single-column model, the internal variability in the model is negligible and no ensembles are necessary. This specific model, SCAM, was designed to evaluate physics parameterizations (Gettleman et al. 2019; https://doi. org/10.1029/2018MS001578). We have modified the paper to clarify this for the reader:

*SCAM has all of the physics parameterizations from the atmospheric component of CESM2, the Community Atmosphere Model Version 6 (CAM), including the radiation scheme RRTMG (Clough et al., 2005; Iacono et al., 2008). SCAM runs the CAM6 physics, including RRTMG, at a single location and prescribes the dynamics state (Gettelman et al., 2019). We forced all SCAM runs with 17 days of observations (temperature and aerosols) from the Mixed-Phase Arctic Cloud Experiment (MPACE) to simulate an Arctic atmosphere with mixed-phase and supercooled liquid-containing clouds (Harrington and Verlinde, 2005).*

What is the motivation for simulations with global models? Such simulations are useful when scales interact - here, if the change in cloud optics might be expected to systematically impact interactions between the Arctic and rest of the world. Is that expected? If not why wouldn't assessment with regional model be more informative?

We ran global models to assess the impact globally since the optics were applied globally. However, we focused on the Arctic because we know optically thin clouds occur there and the optics have the largest effect for those cloud types.

We clarified this important point multiple times in the revised manuscript:

L43-46 revised paper:

*For instance, using a high spectral resolution line-by-line radiative transfer model applied to case studies in the Arctic, Rowe et al. (2013) found that these supercooled liquid water optics can increase modeled longwave fluxes emitted by thin (liquid water path < 10 g m$^{-2}$) supercooled liquid-containing clouds by up to 1.7 W m$^{-2}$.*

L51-53 revised paper:

*We focus on the Arctic because it is a cold and dry region where thin supercooled liquid clouds frequently occur in observations (Cesana et al., 2012) and climate model simulations (McIlhattan et al., 2020). Thus, we anticipate the clouds optics change may have a substantial impact on Arctic longwave fluxes.*

The claim (repeated seven times) that the approach represents a "novel model hierarchy" is not well-founded. "Model hierarchy" refers to sets of equations representing the same underlying system with different levels of complexity. It's a stretch to call a configuration in which winds are relaxed to time-varying empirical values a separate element and the idealized radiative transfer calculations are clearly a different beast. As the authors note the use of wind nudging is not novel. The work can stand on its own without claims to greater generality than are supported.

We very much appreciate and agree with the reviewer's comments. In response, we changed our entire approach to describing the model experiments that we use. Instead in our revised paper, we removed the two-stream radiative transfer model and only present the influence of the cloud optics change in the RRTMG radiation scheme. We then test the influence of the optics change in models with differing levels of dynamical constraint.

See lines 54-60 and section 2.2 of the revised manuscript for an overview of the new approach:

L54-60 revised paper:

*A novel aspect of this study is using a hierarchy of models to assess the relevance of this cloud optics change. All models use the same radiation scheme (RRTMG), but vary in the degree to which the atmosphere is dynamically constrained. We anticipate using this hierarchy of constraint on the modeled atmospheric circulation sequencing will be of value. We expect the most dynamically constrained models will enable the easiest detection of the optics change. In contrast, dynamically unconstrained models will have more noise from internal climate variability and that noise may make it hard to detect the optics change signal. While this study focuses on*

*one specific cloud optics change, the methods used here are applicable to any model physics change and therefore should be of broad interest to the model development community.*

Section 2.2 – L69-83 revised paper:

**2.2 Model hierarchy**

*In this work, we evaluate the effect of changing the liquid water optics from room temperature to supercooled on longwave radiation across a range of dynamically constrained models, while keeping the radiation scheme the same. The models in our hierarchy proceed from the most to least dynamically constrained atmosphere:*

*1. **Single-column atmospheric model:** a completely constrained model at a single location on a daily time scale*

*2. **Wind-nudged global climate model configurations:***

> *(a) **Atmosphere-only (short time scale):** a global dynamically constrained model on an annual time scale*

> *(b) **Atmosphere-only (long time scale):** a global dynamically constrained model on a decadal time scale*

> *(c) **Fully coupled (short time scale):** a global fully coupled dynamically constrained model on an annual time scale*

*3. **Freely evolving global climate model:** an unconstrained global climate model on a decadal time scale*

*For each model, we compared the longwave radiation produced using room temperature water optics against longwave radiation produced using supercooled liquid water optics. Then, we evaluated whether the difference in radiation was detectable and statistically significant. Finally, we assessed at what time and spatial scales and degree of dynamical constraint the supercooled liquid water optics mattered. Primarily, we focus on the downwelling longwave flux at the surface ($W\,m^{-2}$) to evaluate if the optics changed.*

*Some more minor comments*

Most figures could be more carefully crafted to emphasize the narrative points being made. Some figures (2, 12) illustrate concepts that emerge clearly from the text. Others contain information that's visually hard to parse. Figure 3, for examples, requires readers to mentally subtract lines from two different panels, in addition to showing variations with respect to two related by hard-to-interpret quantities, while the information density of figure 4 is low. The authors might fruitfully review each figure and refine those that do not advance the story being told.

We appreciate the reviewer's feedback on these figures. We agree with these concerns. In
response, we have moved Figure 3 to the appendix and changed panels (c) and (d) to be the
difference between the supercooled and current CESM2 optics. We have also removed Figure 4,
agreeing with the reviewer that it adds little to the paper.

[Figure]

*Figure A1. The longwave mass absorption coefficient ($k_{abs}$ (m2 kg−1)) graphed for the current*
*RRTMG liquid optical properties (a) & (b) as function of wavenumber ad wavelength. The*
*difference in longwave mass absorption coefficient between new liquid optical properties*
*calculated from the 263 K complex refractive index (Rowe et al., 2020) and the current RRTMG*
*liquid optical properties (c) & (d) is also graphed as a function of wavenumber and wavelength. In*
*RRTMG, $k_{abs}$ is a lookup table in terms of the parameters μ and 1/λ that describe the droplet size*
*distribution where λ is a function of μ. (b) and (d) are the $k_{abs}$ spectra at a fixed μ and five λ. (a) and*
*(c) are the $k_{abs}$ spectra at five μ and their corresponding λ.*

The captions of Figure 7 and later note that statistical significance is assessed "following Wilks
(2016)" but the text provides no elaboration. Is significance computed accounting for false
discovery rate? If so this should be noted more clearly in the main text.

Yes, we did use Wilks (2016) to account for the false discovery rate. We have added language in
all the relevant figure captions to clarify that for the readers.

One example of an added sentence about Wilks (2016) from the Figure 3 caption in the revised
paper:

*False discovery rate was controlled for using Wilks (2016).*

The authors are quite free with advice to others (e.g. line 238, line 249, line 279). This may be
worth revisiting given the nuanced results obtained.

We agree that the results are nuanced. Detecting a signal due to the cloud optics change required
strong dynamical constraints. When those dynamical constraints were removed, the signal was
not statistically significant at the 95% confidence level above the chaotic atmospheric noise.
Therefore, we think these changes should be added to RRTMG, just not as a first priority.

The formulation of the offline radiative transfer model in appendix A is confusing. The offline
model is used to compute longwave fluxes. Liquid clouds do not scatter longwave radiation so
it's not clear why one would use two-stream equations representing multiple scattering (A7-
A11). It would be far simpler to use Schwartzchild's equation, potentially accounting for intra-
layer temperature gradients as in section 2.1 of Clough et al. 1992 (doi:10.1029/92JD01419).
Indeed that's what models like RRTMG do.

We agree that the offline radiative transfer model is not needed and added unnecessary
confusion. As such, we removed the two-stream radiative transfer model and thus Appendix A1.

Please note, however, that liquid clouds do scatter longwave radiation. We agree that this effect
is very small for downwelling longwave radiation, but multiple scattering by liquid clouds may be
important for upwelling radiation, as it causes biases due to using the incorrect CRI for even the
thickest clouds, as noted by Rowe et al. (2013). While we do not explore upwelling radiation in
this work, since the effect is largest in the tropics and small in the Arctic, we point out in the
discussion that the effect on upwelling longwave radiation is a topic of interest for future work.

The equations in both appendicies are well-known and the tables are available in the original
literature. Since neither sheds light on the problem at hand they can be safely omitted.

We agree with the reviewer about appendix A1. In response, we have removed it.

However, the equations in appendix A2 describing the calculation of CESM2 liquid water optics
were hard to find, understand, and reconstruct from CESM2 CAM6 documentation. Therefore,
we retained these equations in the appendix to make our study reproducible and well
documented.

---

## Author Comment (AC2)

**Response to Reviewer #2 – Gilbert et al. (under review) GMD**

Reviewer comments are in black. Author responses are in blue. *Changes to manuscript are in*
*italic.*

This study investigated the effect of "temperature-dependent cloud optics" on infrared radiation,
with a specific focus on the Arctic region. The analysis is done through a combination of a simple
mathematical model for two-stream radiative transfer, a single-column atmospheric model, an
atmospheric model, and a wind-nudged atmospheric model. The results suggest that the impact
of "temperature-dependent cloud optics" is less significant compared to the internal variability in
the Arctic region. When model winds are nudged towards reanalysis, the internal variability is
partially constrained, and the effect of temperature-dependent cloud optics becomes more
prominent.

This study has the potential to update our understanding of the impact of temperature-dependent
cloud optics on climate simulations. However, there are a few major issues in this manuscript
which I list below. The authors may need to perform additional experiments and data analyses.
And based on that, I would recommend major revision.

We thank the reviewer for their time and constructive review. We provide a point-by-point
response below.

1. It is not correct to claim that the designed model simulations study the effect of
"temperature- dependent cloud optics". The authors simply switched the cloud optics at
298 K in the original model to the cloud optics at other temperatures. It is essentially
cloud optics at a constant temperature (or temperature-independent cloud optics). While
it is OK to simply do this in idealized single-column model experiments, because the
cloud temperature can be set at any value to quantify the flux changes in the extreme
cases, it is not appropriate to do this in the full atmospheric model simulations. Although
the authors mentioned in the discussion section that this will be part of future work,
"temperature-dependent" is still a confusing term to describe the current approach. I
recommend the authors rephrasing it or implementing the physics to the atmospheric
model.

We agree with the reviewer. This issue was also raised by reviewer #1. In response, we
have replaced the term "temperature-dependent" with "supercooled liquid". We have
also replaced "temperature-independent" with "room temperature" (i.e. optics at ~298
K).

2. Based on what has been presented in this manuscript, I don't think the analyses are
sufficiently thorough, and the power of model hierarchy on understanding the impact of
physical assumptions in climate models is not fully realized in this study. For example, the
change of surface downward longwave radiative flux due to the use of temperature-
dependent cloud optics is not well quantified. Only spatial pattern of differences between

| 38 | model runs are shown (Figures 7~10). The ranges given in the manuscript are mostly |
| 39 | approximate (e.g., 1~2 W/m$^2$, 1~3 W/m$^2$, 1~7 W/m$^2$, etc.). Also, only surface downward |
| 40 | longwave flux changes are quantified here, but the impact on OLR is also important from |
| 41 | the perspective of the TOA radiation budget. I suggest that the authors should start from |
| 42 | analyzing the global mean and regional mean time series of OLR and surface downward |
| 43 | longwave flux, providing an estimate of flux differences, and then go further to analyze the |
| 44 | spatial pattern of flux changes. |

| 45 | We agree with the reviewer that we could improve the use of the model hierarchy in the |
| 46 | paper and also improve the quantification. |

| 47 | In response to the first point about the power of the model hierarchy, we restructured the |
| 48 | paper and the model hierarchy as function of dynamical constraint instead of model |
| 49 | complexity. As a part of that restructuring, we have also removed the two-stream |
| 50 | radiative transfer model based on comments from both reviewers and the two-stream |
| 51 | model not fitting within the revised manuscript framing. |

| 52 | In response to the second point about quantification, we have added spatial averages of |
| 53 | the downwelling longwave flux differences. See revised spatial plots and modified Table 3 |
| 54 | for Arctic averages. |

| 55 | With regard to OLR, previous work has shown that the supercooled liquid water optics do |
| 56 | impact downwelling longwave radiation but had little impact in the Arctic on OLR (Rowe |
| 57 | et al. 2013). Similarly, we found very small changes in OLR from the freely evolving |
| 58 | climate model run. In the Arctic, the effect of the supercooled liquid water optics ranged |
| 59 | from a decrease in OLR (0.04 W m$^{-2}$ – 263 K optics) to an increase in OLR (0.23 W m$^{-2}$ – |
| 60 | 273 K optics). Globally, the supercooled liquid water optics increased the OLR 0.08-0.11 |
| 61 | W m$^{-2}$. We added text to the paper but did not add a figure because the effect is small. |

| 62 | L206-211 revised paper: |

| 63 | *Although the results thus far focus on downwelling surface longwave radiation, the* |
| 64 | *supercooled liquid water optics that we implemented impact longwave radiation emitted* |
| 65 | *in all directions. Of critical importance, outgoing longwave radiation emitted at the top of* |
| 66 | *the atmosphere (OLR) contributes to the planetary energy balance. Thus, we also* |
| 67 | *assessed the optics impact on OLR from the freely evolving climate model run. We found* |
| 68 | *the globally averaged OLR changes resulting from the optics changes are small (0.08–* |
| 69 | *0.11 W m$^{-2}$) and not statistically significant. Thus, this short analysis of the OLR provides* |
| 70 | *additional evidence that the influence of the optics change on the freely evolving model is* |
| 71 | *modest.* |

| 72 | Finally, we elected to not add timeseries of the fluxes. We think the maps and spatial |
| 73 | averages provide ample information to assess the influence of our changes on the mean |

state. The results are small, and as such, investigating variability seems of second order
importance.

3. For the two-stream radiative transfer model described in section 2.3, the authors chose
to use a very simple mathematical model to do the calculation. This does not take into
account the atmospheric absorption, while it is an important factor that may mask the
effect of cloud optics change. The authors may use a more developed two-stream
radiative transfer model. For example, RRTMG_LW provides a single-column version that
users can specify any profile to test. Using this model, the authors can calculate the flux
differences in broad cases and even plot the sensitivity of flux difference to the
meteorological factors and cloud properties.

We agree with the reviewer. This point was also brought up by reviewer #1. In response,
we removed the two-stream radiative transfer model from the revised paper.

4. For the single-column atmospheric model, what variables are prescribed by the
observations? My understanding is that clouds are not constrained by the observations.
For most observational period in Figure 6, the flux difference is very close to 0. Are they
cloud-free scenarios? I would suggest filtering out the clear-sky cases and focus on the
cloudy scene.

Here, we clarify the specific variables used to force the single-column atmospheric
model. The variables the model relaxed to were observations of temperature and aerosols
at every vertical level. The specific variable names listed in the SCAM code were 'T',
'bc_a1', 'bc_a4', 'dst_a1', 'dst_a2', 'dst_a3', 'ncl_a1', 'ncl_a2', 'ncl_a3', 'num_a1',
'num_a2', 'num_a3', 'num_a4', 'pom_a1', 'pom_a4', 'so4_a1', 'so4_a2', 'so4_a3', 'soa_a1',
and 'soa_a2', also available in the SCAM namelists we provided. As for the second point
about filtering out clear-sky scenes, there were none for the period modeled by the SCAM.
This information is provided in Gettelman et al. 2019 and in the SCAM documentation.
Thus we do not repeat the specific variables in our paper. Instead, we state generally what
is used to force SCAM and point the reader to this paper describing SCAM.

L93-98 revised paper:

*SCAM has all of the physics parameterizations from the atmospheric component of*
*CESM2, the Community Atmosphere Model Version 6 (CAM), including the radiation*
*scheme RRTMG (Clough et al., 2005; Iacono et al., 2008). SCAM runs the CAM6 physics,*
*including RRTMG, at a single location and prescribes the dynamics state (Gettelman et*
*al., 2019). We forced all SCAM runs with 17 days of observations (temperature and*
*aerosols) from the Mixed-Phase Arctic Cloud Experiment (MPACE) to simulate an Arctic*
*atmosphere with mixed-phase and supercooled liquid-containing clouds (Harrington and*
*Verlinde, 2005).*

**Specific Comments**

1. L19-21: A reference may be necessary to support the statement that "All else being equal, clouds with small particle sizes also scatter more shortwave and emit more downwelling longwave than clouds with large particle sizes.

   We agree with the reviewer's suggestion. In response, we have added Maahn et al. 2021 (https://doi.org/10.1029/2021GL094307) to support the statement "clouds with small particle sizes also scatter more shortwave" and Lubin and Vogelmann 2006 (https://doi.org/10.1038/nature04449) to support the statement "emit more downwelling longwave".

   L19-21 revised paper:

   *All else being equal, clouds with small particle sizes also scatter more shortwave (Maahn et al., 2021) and emit more downwelling longwave than clouds with large particle sizes (Lubin and Vogelmann, 2006).*

2. L39-40: "Specifically, temperature-dependent liquid water optics are not used in RRTMG." Related to the first major issue, this sentence is very confusing as the authors did not implement the full temperature-dependent liquid water optics in the model, either. The authors may be more specific on what specific cloud optics RRTMG has used (e.g., at 298 K), and point out that this may not reflect the truth in the supercooled liquid cloud regime.

   We agree with the reviewer. In response we have modified the sentence. We changed "temperature-dependent" to "supercooled liquid" and have added sentences to make the reviewer's last point.

   L41-43 revised paper:

   *Specifically, supercooled liquid water (240–273 K) optics are not used in RRTMG. Instead, RRTMG uses liquid water optics at one fixed temperature (298 K). Since the RRTMG optics temperature doesn't match supercooled liquid cloud temperatures, the RRTMG optics may not represent radiation emitted by supercooled liquid-containing clouds well.*

3. L39: Also cite Clough et al. (2005; https://doi.org/10.1016/j.jqsrt.2004.05.058)

   We agree. In response, we have added the citation to Clough et al. 2005 as suggested by the reviewer.

   L39-41 revised paper:

*We identify a cloud optics physics that has not been incorporated into the radiation scheme used by many climate models, RRTMG (Clough et al., 2005; Iacono et al., 2008).*

4. L45-47: This long sentence is a bit confusing. "supercooled liquid clouds frequently occur in both observations [...] and the climate model [...] and where the atmosphere is typically cold and dry." These three are not in parallel. Consider this alternative: "supercooled liquid clouds frequently occur in the cold and dry region, as evidenced by observations and climate model simulations."

We agree. In response, we have substituted the reviewer's phrasing in the paper.

L51-52 revised paper:

*We focus on the Arctic because it is a cold and dry region where thin supercooled liquid clouds frequently occur in observations (Cesana et al., 2012) and climate model simulations (McIlhattan et al., 2020).*

5. L92: For surface, "albedo" is specific for solar radiation. A better term could be "reflectivity".

We agree. That said, we removed the entire two-stream radiative transfer model section so this change is no longer relevant.

6. Figure 2: "reflected ground emission" is ambiguous. A better alternative is "ground emission scattered by clouds"

We agree. That said, we removed the entire two-stream radiative transfer model section so this change is no longer relevant.

7. Figure 2: In longwave radiative transfer, it better aligns with the convention to use emissivity rather than reflectivity.

We agree. That said, we removed the entire two-stream radiative transfer model section so this change is no longer relevant.

8. Figure 3: For panels (c) and (d), it could be better to visualize the difference between 263 K optics and CESM optics.

We agree. In response, we have modified Figure 3(c) & (d) to show the difference between the 263 K and CESM control optics. We have also moved Figure 3 to appendix A, so it is now labeled Figure A1, as we think that it fits better there than in the main body of the paper.

[Figure]

*Figure A1. The longwave mass absorption coefficient ($k_{abs}$ (m2 kg−1)) graphed for the current RRTMG liquid optical properties (a) & (b) as function of wavenumber ad wavelength. The difference in longwave mass absorption coefficient between new liquid optical properties calculated from the 263 K complex refractive index (Rowe et al., 2020) and the current RRTMG liquid optical properties (c) & (d) is also graphed as a function of wavenumber and wavelength. In RRTMG, $k_{abs}$ is a lookup table in terms of the parameters $\mu$ and $1/\lambda$ that describe the droplet size distribution where $\lambda$ is a function of $\mu$. (b) and (d) are the $k_{abs}$ spectra at a fixed $\mu$ and five $\lambda$. (a) and (c) are the $k_{abs}$ spectra at five $\mu$ and their corresponding $\lambda$.*

9. Table 1: Do these model runs include model spin-up period? It takes time for the model to adjust to the new state.

   Analysis of timeseries showed little evidence for a need to spin-up the model. When the atmosphere is freely evolving, atmospheric processes spin up within days. When the wind nudging is being used, spin up is not a concern for this work.

10. Table 1: Why is the 263 K run missing in the F1850 experiment? Especially consider that Figure 3 highlights the comparison between 263 K optics and CESM optics, and also the 263 K run appears in all other experiments.

    Initially we wanted to test the extremes of optics set, 240 K and 273 K, and so we only ran those optics sets for the F1850 experiment. After those experiments, we evaluated which optics set was the closest to Arctic cloud temperature and found that 263 K was the closest. Thereafter we used 263 K optics.

All this said - we agree with the reviewer that considering all other experiments have a 263 K optics run, F1850 should as well. In response, we ran and added an F1850 263 K optics run. While this addition does make the study more complete, it did not change the main results.

11. L141: "the next time step". Note that 6-hourly ERA-Interim reanalysis is used here while the model step is 30 minutes by default. According to the referred literature, this is indeed the next available analysis time, not the next model time. Please be more specific and clear.

We agree. In response, we have clarified our language in this sentence.

L114-120 revised paper:

*Nudging is implemented following:*

*... (Equations)*

*where F (x) the internal tendency without nudging, $F_{nudge}$ is the nudging term, α is the strength coefficient that is 0 where nudging is not enabled and 1 where nudging is enabled, O(t'next) is the target state at future target time step, x(t) is the model state at the current model time step, and τ is the relaxation time between the next target time step and the current model time step (Blanchard-Wrigglesworth et al., 2021; Roach and Blanchard-Wrigglesworth, 2022).*

12. Figure 4: In panel (a), I noticed that there is a smoothing gradient at the boundaries of the latitudinal band. The previous study cited by the authors explicitly mentioned that they applied smoothing (by setting $\alpha$ to a value between 0 and 1 in some region). Did the authors also apply the same technique? Also, in panel (b), a solid line is connected between $\alpha$ = 0 and $\alpha$ = 1 at around 800 hPa. Is the smoothing technique also applied here? To make it clear, instead of using line plot, the authors may choose scatter plot instead to visualize the exact $\alpha$ values at each discrete layer.

Yes, the authors smoothed both at the vertical boundary and horizontal boundary using a sharpness parameter provided in the nudging namelists. We have added a sentence to clarify this for the reader.

L121-122 revised paper:

*At both the vertical and horizontal nudging boundaries, we applied smoothing.*

13. L164-165: "the downwelling irradiance and flux was higher for temperature-dependent optics than temperature-independent optics" This is confusing. It would be better to state that the downwelling irradiance and flux was higher for cloud optics at X temperature than
the optics at Y temperature.

We agree. That said, we removed the entire two-stream radiative transfer model section
so this change is no longer relevant.

14. L165: "The thinnest clouds […] showed the largest difference." This statement is not
supported by Figure 5, as no results are presented for clouds at different thickness.

We agree. That said, we removed the entire two-stream radiative transfer model section
so this change is no longer relevant.

15. L167: What is the meaning of "all cloud temperatures"? Rephrase this sentence.

We agree. That said, we removed the entire two-stream radiative transfer model section
so this change is no longer relevant.

16. L168-169: "However, as cloud thickness increased from 100 to 500 m […]" This is not
shown in any figure.

We agree. That said, we removed the entire two-stream radiative transfer model section
so this change is no longer relevant.

17. L170~171: "but our model was meant to be a proof of concept and not realistic". Why not
use a realistic model, given that a quantitative estimate of the effect is provided above
(0.35 W/m$^2$)?

We agree. That said, we removed the entire two-stream radiative transfer model section
so this change is no longer relevant. Additionally, previous work had used a high resolution
line-by-line radiative transfer model to demonstrate an effect from the optics of 1.7 W m$^{-2}$
(Rowe et al. 2013).

18. L177-179: The authors mentioned that when cloud optics at different temperatures are
used, the cloud fraction and cloud phase in the simulations are different. I assume that
the authors do not prescribe the model simulations with observed clouds. What are the
differences in cloud fraction and properties exactly? Having these differences, I don't
think this is an apple-to-appple comparison to show the net effect of cloud optics at
different temperatures since cloud variability has played a role.

We plotted the differences in cloud fraction, cloud liquid, cloud ice, and dominant cloud
species between all the SCAM runs. We found little difference in all cloud properties
between the optics sets. However, those differences in cloud properties concurrently
occurred with the large differences (over 10 W m-2) between the different optics SCAM
runs. These large differences also drove our decision to subset the downwelling longwave fluxes, only including optically thin low-level supercooled liquid clouds. This subsetting
removed any large flux differences caused by cloud property and phase differences.

19. Figure 7: I don't see stippling in the figure, so it is better to say that no significance in the
figure caption.

We appreciate this suggestion, but did not add it. We think it is clearer to state the
significant results would be stippled. We do not want text that could be confusing saying
the double negative of results that are not significant are not stippled.

20. Figure 8: What's the regional mean difference in these plots? The average can be
performed over 50oN~90oN, consistent with the given latitudinal band in Table 3, and the
values can be added to the panel title.

We agree and have added the regional mean difference to Table 3. However, we
calculated the regional mean over 60-90N to match the wind nudging domain.

21. L208: I suggest adding "at 5% significance level" to be more accurate and specific.

We agree. In response, we have fixed the language.

L172-173 revised paper:

*Critically, many flux differences were statistically significant at the 95 % confidence level.*

22. L209-210: "because the wind nudging reduced the variability in the annual mean flux
between the ensemble members" A figure may be necessary to show this. If there are too
many figures, consider combining the information in one figure. For instance, Figures 7~10
show similar information and can be merged into one figure.

This sentence was removed in our revised manuscript. In response, we re-worded:

L174-176 revised paper:

*The flux differences were statistically significant in this experiment because the wind*
*nudging reduced noise caused by different atmospheric circulation sequences and*
*emphasized the signal from the supercooled liquid water optics.*

Second, wind nudging has been shown in prior studies to reduce ensemble spread
between members (Roach and Blanchard-Wrigglesworth 2022).

[Figure]

The figure shown above is from Roach & Blanchard-Wrigglesworth 2022 (Fig. 2a) and
plots Arctic surface temperature from observations (OBS, black), the CESM1 large
ensemble (LENS, blue), and a wind-nudged ensemble (aNUDGE, red). The 40 CESM1
large ensemble members are dynamically unconstrained and have considerable
ensemble spread between members. Additionally, the members do not sync up with the
interannual variability of the observations. However, all nudged ensemble members
match very closely with each other and observations, substantially reducing spread
between members. This example shows that nudging all ensemble members to the same
set of winds can reduce spread and variability between the members.

23. L218~220: "no flux differences [...] were statistically significant" Instead of setting some
threshold, I suggest providing a $p$-value so that we can understand how far it is from the
significance threshold.

For our method of controlling the false discovery rate (Wilks 2016), the critical threshold
(normally 0.05), is modified as a function of the p-values.  In the case of this experiment,
the value for the critical threshold revealed that no flux differences had statistically
significant p-values. Additionally, each grid box has its own p-value, so there is no single
p-value to provide.

24. L232-236: Given that the authors simply change the cloud optics at another temperature,
the effect on mean 2-m air temperature difference should be more prominent than the
effect on 2-m air temperature trend, since the temperature-cloud property feedback is
muted. Also, considering that no greenhouse gas and aerosol forcings are included in the
simulations, it makes no sense to compare to the ERA-I 2-m air temperature trend.

We agree with the reviewer. We also concluded that the temperature time series doesn't
make sense to include in this paper. In response, we removed the figure and any
discussion of surface temperature.

25. L246-247: "Whereas for the global cliate model, an effect of a few W m$^{-2}$ is within climate
variability and thus relatively small." Note that the historical change in effective radiative
forcing from 1750 and 2019 is also a few W m$^{-2}$.

The reviewer is correct that the results are nuanced. As we wrote to reviewer #1,
detecting a signal due to the cloud optics change required strong dynamical constraints.
When those dynamical constraints were removed, the signal was not statistically
significant at the 95% confidence level above the chaotic atmospheric noise. Additionally
here, we note that the effect is small (less than 1 W m$^{-2}$) and also smaller than the
observed change in effective radiative forcing.

26. Table 3: The values in the "Effect of optics" column should be the regional mean values as
defined in the "Spatial scale" column.

We agree. In response, we have added the spatial mean values as defined in the spatial
scale column to Table 3. Any value ranges in the "Effect of optics" column represent the
minimum and maximum effects from multiple supercooled optics sets, i.e. 0.36-0.68 W
m$^{-2}$ where 0.36 is the effect of the 263 K optics and 0.68 is the effect of the 273 K optics.
The spatial mean differences have also been added to all spatial plots.